# GLC_FCS30: Global land-cover product with fine classification system at 30 m using time-series Landsat imagery

Xiao Zhang [1], Liangyun Liu [1, 2], Xidong Chen[1, 2], Yuan Gao[1, 3], Shuai Xie[1, 2] and Jun Mi[1, 2]

[1] State Key Laboratory of Remote Sensing, Aerospace Information Research Institute, Chinese Academy of Sciences, Beijing 100094, China

[2] University of Chinese Academy of Sciences, Beijing 100049, China University of Chinese Academy of Sciences, Beijing 100049, China

[3] College of Geomatics, Xi'an University of Science and Technology, Xi'an 710054, China

*Correspondence to*: Liangyun Liu (liuly@radi.ac.cn)

**Abstract.** Over past decades, a lot of global land-cover products have been released, however, these is still lack of a global land-cover map with fine classification system and spatial resolution simultaneously. In this study, a novel global 30-m land-cover classification with a fine classification system for the year 2015 (GLC_FCS30-2015) was produced by combining time-series of Landsat imagery and high-quality training data from the GSPECLib (Global Spatial Temporal Spectra Library) on the Google Earth Engine computing platform. First, the global training data from the GSPECLib were developed by applying a series of rigorous filters to the CCI_LC (Climate Change Initiative Global Land Cover) land-cover and MCD43A4 NBAR products (MODIS Nadir Bidirectional reflectance distribution function-adjusted Reflectance). Secondly, a local adaptive random forest model was built for each 5°×5° geographical tile by using the multi-temporal Landsat spectral and textures features and the corresponding training data, and the GLC_FCS30-2015 land-cover product containing 30 land-cover types was generated for each tile. Lastly, the GLC_FCS30-2015 was validated using three different validation systems (containing different land-cover details) using 44,043 validation samples. The validation results indicated that the GLC_FCS30-2015 achieved an overall accuracy of 82.5% and a kappa coefficient of 0.784 for the level-0 validation system (9 basic land-cover types), an overall accuracy of 71.4% and kappa coefficient of 0.686 for the UN-LCCS (United Nations Land Cover Classification System) level-1 system (16 LCCS land-cover types), and an overall accuracy of 68.7% and kappa coefficient of 0.662 for the UN-LCCS level-2 system (24 fine land-cover types). The comparisons against other land-cover products (CCI_LC, MCD12Q1, FROM_GLC and GlobeLand30) indicated that GLC_FCS30-2015 provides more spatial details than CCI_LC-2015 and MCD12Q1-2015 and a greater diversity of land-cover types than FROM_GLC-2015 and GlobeLand30-2010, and that GLC_FCS30-2015 achieved the best overall accuracy of 82.5% against FROM_GLC-2015 of 59.1% and GlobeLand30-2010 of 75.9%. Therefore, it is concluded that the GLC_FCS30-2015 product is the first global land-cover dataset that provides a fine classification system (containing 16 global LCCS land-cover types as well as 14 detailed and regional land-cover types) with high classification accuracy at 30 m. The GLC_FCS30-2015 global land-cover products produced in this paper is free access at https://doi.org/10.5281/zenodo.3986871 (Liu et al., 2020).

# 1 Introduction

Global land-cover information, as used by the scientific community, governments and international organizations, is critical to the understanding of environmental changes, food security, conservation and the coordination of actions needed to mitigate and adapt to global change (Ban et al., 2015; Chen et al., 2015; Tsendbazar et al., 2015). These data also play an important role in improving the performance of models of the ecosystem, hydrology and atmosphere (Gong et al., 2013). Accurate and reliable information on global land cover is, therefore, urgently needed (Ban et al., 2015; Zhang et al., 2019).

Due to the frequent and large-area coverage that it provides, more and more attention has been attached to using the remote sensing technology for global land-cover mapping. In past decades, several global land-cover products have been produced at various spatial resolutions ranging from 1 km to 300 m (Bontemps et al., 2010; Defourny et al., 2018; Friedl et al., 2010; Loveland et al., 2000; Tateishi et al., 2014). However, owing to differences in classification accuracy, thematic detail, classification schemes, and spatial resolution, the harmonization of these land-cover products is usually difficult (Ban et al., 2015; Gómez et al., 2016; Giri et al., 2013; Grekousis et al., 2015) and their quality is also far from satisfactory for many fine applications (Giri et al., 2005; Grekousis et al., 2015; Yang et al., 2017b). Recently, thanks to free access to fine-resolution remote sensing imagery (Landsat and Sentinel-2), combined with rapidly increasing data-storage and computation capabilities, global land-cover products at fine spatial resolutions (10 m and 30 m) have been successfully developed (Chen et al., 2015; Gong et al., 2019; Gong et al., 2013). Specifically, Chen et al. (2015) used multi-temporal Landsat and similar image data along with the integration of pixel- and object-based methods to produce the GlobeLand30 land-cover product that has an overall classification accuracy of over 80%. Similarly, Gong et al. (2013) and Gong et al. (2019) produced the global 30-m and 10-m land-cover products (FROM_GLC30 and FROM_GLC10) using single-date Landsat imagery and multi-temporal Sentinel-2 imagery, respectively. Unlike FROM_GLC10 and GlobeLand30, which have only 10 land-cover types, FROM_GLC30 was classified using 28 detailed land-cover types. However, as the overall accuracy for the detailed land-cover types was only 52.76% and the patch effects was noticeable caused by the temporal differences among the Landsat scenes, FROM_GLC30 focused on the mapping results for just 10 major land-cover types (Gong et al., 2013). Although these products permit the detection of land information at the scale of most human activity and offer increased flexibility for the environmental model parameterization needed for global land-cover studies (Ban et al., 2015), the simple classification system and large amount of manual work required (manual collection of training samples and knowledge-based interactive verification) limit their greater use in many specific and fine applications at regional or global scales.

As Giri et al. (2013) and Ban et al. (2015) stated that there are a number of challenges to overcome in producing a fine-resolution characterization of global land cover. These include the unavailability of timely, accurate and sufficient training data, the high cost of collecting satellite data with consistent global coverage, difficulties in preparing image mosaics, as well as the need for high-performance computing facilities.

Firstly, Foody and Arora (2010) stated that the training data had more impact on the classification results than the selection of the classifier: the collection of timely, accurate and sufficient training data are especially important for global or regional land-

cover mapping. Generally, the collection of training data can be divided into two types of method: interpretation-based methods, and the derivation of training samples from existing land-cover products. Specifically, the interpretation-based methods are widely used in regional land-cover classification because high confidence in the training data can be guaranteed (Xie et al., 2018; Zhu et al., 2016). However, for large-area land-cover mapping, the interpretation of sufficient and accurate training data usually involves a huge amount of manual work. For example, Gong et al. (2013) collected 91,433 training

samples using 27 image analysts who were experienced in remote-sensing image interpretation. Similarly, Tateishi et al. (2014) selected 312,753 training points from 2,080 prior training polygons (Tateishi et al., 2011) and used a large amount of reference data, including Google Earth images from around 2008, existing regional land-cover maps, and MODIS NDVI phenological curves from 2008. Despite the total number of training samples apparently being large in the works of Gong et al. (2013) and Tateishi et al. (2014), in fact, in terms of global land-cover mapping, these training samples still provided only sparse coverage:

Zhu et al. (2016) suggested that the optimal number of training pixels needed to classify an area about the size of a Landsat scene was about 20,000. Furthermore, the land-cover diversity (the number of land-cover types in the final results) of training data is also constrained by the available expert knowledge: for example, Chen et al. (2015) produced a global land-cover product (GlobeLand30) containing only 10 land-cover types; Gong et al. (2019) developed the first global 10-m land-cover product (FROM_GLC10), which also contained 10 major land-cover types.

Compared with the interpretation-based methods, the second type of data collection method – deriving training samples from existing land-cover products – has been demonstrated to have many significant advantages, including fully automated collection and refinement of training data, the production of a large and geographically distributed training dataset, and the possibility of using the same land-cover classes as existing land-cover products (Inglada et al., 2017; Jokar Arsanjani et al., 2016b; Liu et al., 2017; Radoux et al., 2014; Wessels et al., 2016; Xian et al., 2009; Zhang and Roy, 2017; Zhang et al., 2019;

Zhang et al., 2018). For these reasons, this type of data collection has recently attracted more attention in large-area land-cover mapping. For example, Radoux et al. (2014) used the coarse resolution land-cover products, Global Land Cover (GLC) 2000 and Corine Land Cover (CLC) 2006, to develop 300-m land-cover results for South America and Eurasia respectively; Zhang and Roy (2017) used the MODIS land-cover product (MCD12Q1) to classify time-series of Landsat imagery and then produce a 30 m land-cover classification of north America, achieving an overall agreement of 95.44% and a kappa coefficient of 0.9443.

Recently, Zhang et al. (2019) proposed simultaneously using the MODIS Nadir bidirectional reflectance distribution function-adjusted reflectance (MCD43A4 NBAR) and the CCI_LC (European Space Agency Climate Change Initiative Global Land Cover) land-cover product from 2015 to generate a 30-m Landsat land-cover dataset for China. However, as well as these advantages, there is the problem that the derived training data might be affected by classification errors in the existing land-cover products and by spatial resolution and temporal differences between the land-cover products and the satellite data that

are to be classified. In recent years, many researchers have proposed various measures to ensure that only reliably defined training data are extracted: for example, Radoux et al. (2014) proposed the use of spatial and spectral filters to remove outliers, Zhang and Roy (2017) proposed that only MCD12Q1 pixels that had been stable for three consecutive years should be used and that these pixels should be refined using the "metric centroid" method developed by Roy and Kumar (2016). In summary,

if effective measures can be taken to control the confidence and reliability of the training data, the derivation of training samples from existing land-cover products has great potential for global land-cover mapping.

Secondly, the high cost of collecting satellite data with consistent global coverage, the lack of high-performance computing requirements and the difficulties in preparing image mosaics also cause problems. However, because the Google Earth Engine (GEE) cloud-based platform consists of a multi-petabyte analysis-ready data catalog co-located with a high-performance, intrinsically parallel computation service, and because the library's image-based functions in the GEE are per-pixel algebraic operations (Gorelick et al., 2017), these difficulties can be easily solved by using the GEE cloud-computation platform. In recent years, many large-area land-cover classifications have been produced based on the GEE cloud computation platform: for example, Teluguntla et al. (2018) successfully derived 30-m cropland extent products for Australia and China, which had overall accuracies of 97.6% and 94%, on the GEE platform. Gong et al. (2019) produced the first global 10-m land-cover product using time-series of Sentinel-2 imagery also on the GEE platform.

Overall, due to the difficulties in collecting sufficient accurate training data with a fine classification system and the computing requirements involved, producing a global 30-m land-cover classification with a fine classification system is a challenging and labor-intensive task. This paper presents an automatic classification strategy for producing a global land-cover product with a fine classification system at a spatial resolution of 30 m for 2015 (GLC_FCS30-2015) using the Google Earth Engine cloud computation platform. To achieve this goal, we first derived the global training data from the updated Global Spatial Temporal Spectra Library (GSPECLib), which was developed by combining the MCD43A4 NBAR surface reflectance product and the CCI_LC land-cover product for 2015. Secondly, time-series of Landsat imagery on the GEE platform were collected and then temporally composited into several temporal spectral and texture metrics using the metrics-composite method. Finally, by combining a multi-temporal random forest model, global training data and Landsat temporal features, a global annual land-cover map with 30 land-cover types was produced. The validation results indicated that the GLC_FCS30-2015 is a promising land-cover product and could provide important support for numerous regional or global applications.

## 2 Datasets

### 2.1 Satellite datasets

#### 2.1.1 Landsat surface reflectance data

Taking account of the frequent contamination of cloud in the remote sensing imagery, particularly in the tropics, all Landsat-8 surface reflectance (SR) imagery from 2014–2016 archived on the GEE platform was collected for the nominal year 2015. Each Landsat-8 SR image on the GEE was atmospherically corrected by the Landsat Surface Reflectance Code (LaSRC) atmospheric correction method (Roy et al., 2014; Vermote et al., 2016), and bad pixels – including cloud, cloud shadow and saturated pixels – were identified by the CFMask algorithm (Zhu et al., 2015; Zhu and Woodcock, 2012). In this study, only

six optical bands – blue, green, red, NIR, SWIR1 and SWIR2 – were used for land-cover classification because the coastal
band is easily effected by the atmosphere conditions (Wang et al., 2016).

Fig. 1 illustrates the clear-sky Landsat-8 SR temporal frequency after the cloud, cloud shadow and saturated pixels have been
masked out. The statistical results indicated that: 1) most land areas, except for tropical areas, had a high availability of clear-
sky Landsat imagery; and2) areas with a low frequency of clear-sky Landsat SR were mainly located in rainforest areas
including the Amazon rainforest, African rainforests and Indian–Malay rainforests, which are areas mainly covered by
evergreen broadleaved forests.

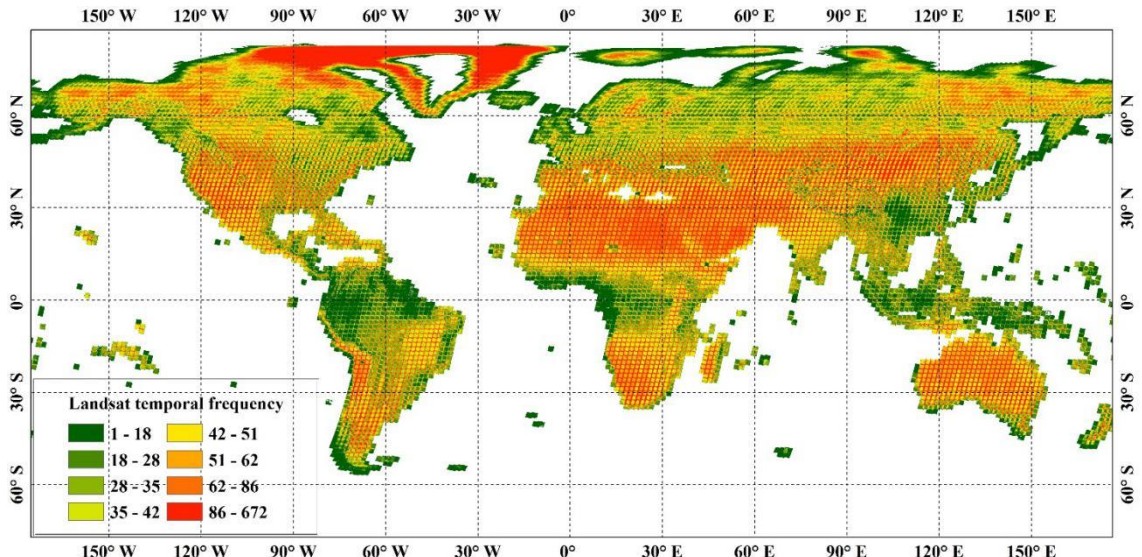

**Figure 1: The availability of clear-sky Landsat SR imagery for the years 2014–2016 on the GEE platform.**

### 2.1.2 Digital elevation model data

Over the past few years, many studies have demonstrated that a digital elevation model (DEM) and variables derived from it
(slope and aspect) are necessary and important auxiliary variables for land-cover mapping (Gomariz-Castillo et al., 2017;
Zhang et al., 2019). In this study, the Shuttle Radar Topography Mission (SRTM) DEM, which has a spatial resolution of 30
m and covers the area between 60° north and 56° south (Farr et al., 2007), and the slope and aspect variables, were used as the
classification features. It should be noted that this dataset archived on the GEE platform has been optimized by a void-filling
process that uses other open-source DEM data. Furthermore, to complement the missing SRTM data at high latitudes, the
GDEM2 DEM dataset (Tachikawa et al., 2011) was collected.

### 2.2 Global 30-m impervious surface products

Due to the spectral heterogeneity and complicated make-up of impervious surfaces, large-area impervious mapping is usually
challenging and difficult (Chen et al., 2015; Gong et al., 2013; Zhang and Roy, 2017). For example, in our previous work

Zhang et al. (2019), impervious surfaces had a low producer's accuracy of 50.7% because fragmented impervious surfaces such as rural cottages, roads etc. were easily missed. Therefore, Chen et al. (2015) split the impervious surface class into three independent sub-classes including 'vegetated', 'low reflectance' and 'high reflectance', and then used the classification method of integrating pixel- and object-based techniques and manual editing to produce accurate global impervious surface products. In this study, the global land-cover classification neglected the impervious surface land-cover type when building the classification model; instead, existing global 30-m impervious surface products for 2015 (MSMT_IS30-2015) were directly superimposed over the global land-cover classifications (Zhang et al., 2020). The MSMT_IS30-2015 dataset was produced in our previous work and developed by combining 420,000 Landsat-8 SR and 83,500 Sentinel-1 SAR images from around the globe on the GEE platform. The validation results indicated that the MSMT_IS30-2015 product achieved an overall accuracy of 95.1% and a kappa coefficient of 0.898 using 11,942 validation samples from fifteen representative regions. The MSMT_IS30-2015 dataset is available at https://doi.org/10.5281/zenodo.3505079 (Zhang and Liu, 2019).

## 2.3 Global validation datasets

To guarantee the confidence of the validation points, several existing prior datasets (see Table 1), high-resolution Google Earth imagery and time-series of NDVI values for each vegetated point were integrated to derive the global validation datasets. Many studies have demonstrated that inappropriately sized validation sample could lead to limited and sometimes erroneous assessments of accuracy (Foody et al. 2009 and Olofsson et al. 2014), therefore, a stratified random sampling based on the proportion of the land-cover areas was adapted to determine the sample size of each land-cover type:

$$n_i = n \times \frac{W_i \times p_i(1-p_i)}{\sum W_i \times p_i(1-p_i)}; \quad n = \frac{(\sum W_i \times \sqrt{S_i(1-S_i)})^2}{[S(\hat{O})^2 + \sum W_i \times S_i(1-S_i)/N]} \approx \left(\frac{\sum W_i S_i}{S(\hat{O})}\right)^2 \quad (1)$$

where $W_i$ was the area proportion for class $i$ over the globe, $S_i$ is the standard deviation of class $i$, $S(\hat{O})$ is the standard error of the estimated overall accuracy, $p_i$ is the expected accuracy of class $i$ and $n_i$ represents the sample size of the class $i$.

First, the cropland-related validation samples were directly inherited from the Global Cropland reference data, which were first collected by worldwide crowdsourcing using the ground data-collection mobile app and then reviewed using high-resolution imagery in the online image-interpretation tool to ensure that the samples were centered on agricultural fields. There are 22,823 cropland validation samples in the reference dataset (Xiong et al., 2017). In addition, due to the possible temporal interval between the acquisition of the reference data and the GLC_FCS30 products (2015), the reference samples were checked by three interpreters using the high-resolution imagery for 2015 in the Google Earth software, and were discarded if the judgements of three experts were in disagreement. After discarding wrong cropland points and resampling using the formula (1), a total of 6,917 cropland samples in 2015 were retained.

Secondly, the GOFC_GOLD datasets contained several reference datasets which included: the Global Land Cover National Mapping Organizations (GLCNMO) 2008 training dataset, the VIIRS land-cover product Visible Infrared Imaging Radiometer Suite (VIIRS) dataset, the MODIS Land Cover (MCD12Q1) product System for Terrestrial Ecosystem Parameterization (STEP) dataset, the GlobCover2005 validation database, and the GLC2000 database (Herold et al., 2010). In this study, the

GlobCover2005 and GLC2000 datasets were removed because they were too sparse and also because the temporal difference between them and our GLC_FCS30-2015 products was too big. The GLCNMO, VIIRS and STEP datasets all contained numerous validation polygons, so we first rechecked each validation polygon against the high resolution imagery for 2015 and then randomly selected several validation points within each refined polygon.

Specifically, as the GLCNMO used the UN LCCS (United Nations Land Cover Classification System), similar to our study (Table 2), and the VIIRS and STEP datasets followed the IGBP (International Geosphere Biosphere Programme) classification system, and as the land-cover types had consistent definitions in both the UN LCCS and IGBP classification systems (including land-cover ids 50, 60, 70, 80, 90, 130 and 200: see Table 2), the corresponding validation points were randomly collected from each polygon for all three datasets. For other land-cover types, where there were slight differences according to the two

classification systems (120 and 150), the validation points were selected from within the GLCNMO polygons only.

Thirdly, the FROM_GLC validation dataset was only used to complement our validation datasets because of the discrepancy between the classification systems (Li et al., 2017a). The lichens and mosses land-cover type (140) was missing in the GOFC_GOLD datasets, the shrubland polygons (120) in the GLCNMO dataset were too sparse, and the impervious surface polygons (190) in GOFC_GOLD were not suitable for validation of the impervious surfaces at a resolution of 30 m because

the impervious surfaces within the polygons were usually broken and heterogeneous. Therefore, the shrubland, tundra and impervious samples in the FROM_GLC validation dataset were collected and then refined using the high-resolution imagery for 2015.

Afterwards, the GLWD dataset, which had a spatial resolution of 30 arcsec and contained 12 lake and wetland classes (Lehner and Döll, 2004; Tootchi et al., 2019), was used to derive the validation samples for the water body (210) and wetland (180) classes. To further ensure confidence in these validation samples, they were rechecked by the interpreters using high-resolution

Google Earth imagery for the year 2015.

The time series of NDVI (Normalized Difference Vegetation index) values for each validation point, derived from the Landsat SR imagery time series, were used to help distinguish between the vegetation-related land-cover types, for example, evergreen shrubland (121) and deciduous shrubland (122), evergreen broadleaved/needleleaved forests (50, 70), and deciduous

broadleaved/needleleaved forests (60, 80).

Lastly, as the ice and snow cover generally varied with time, the time-series of NDSI (Normalized Difference Snow Index) values and high-resolution imagery were combined to collect high-confidence permanent ice and snow (220) samples. Overall, after the combination of the auxiliary datasets from multiple sources and careful rechecking by several interpreters, a total of 44,043 validation samples for 24 fine land-cover types were finally collected – see Fig. 2. The global validation dataset is

publicly available at http://doi.org/10.5281/zenodo.3551994 (Liu et al., 2019).

**Table 1. Multi-source auxiliary datasets used for collecting the global validation samples**

| Dataset name | Target land-cover id |
| --- | --- |
| Global Cropland reference data<br>https://croplands.org/app/data/search?page=1&page_size=200 | 10, 11, 12, 20 |

| | |
|---|---|
| Global Observation for Forest Cover and Land Dynamics (GOFC_GOLD) reference data http://www.gofcgold.wur.nl/sites/gofcgold_refdataportal.php | 50, 60, 70, 80, 90, 120, 121, 122, 130, 150, 152, 153, 200, 201, 202 |
| FROM_GLC global validation sample set http://data.ess.tsinghua.edu.cn | 120, 121, 122, 140, 190 |
| Global Lakes and Wetlands Database (GLWD) https://www.worldwildlife.org/pages/global-lakes-and-wetlands-database | 180, 210 |
| NDVI time-series datasets | 50, 60, 70, 80, 120, 121, 122 |
| NDSI time-series datasets | 220 |

**Note**: For details of the land-cover ids refer to **Table 2**

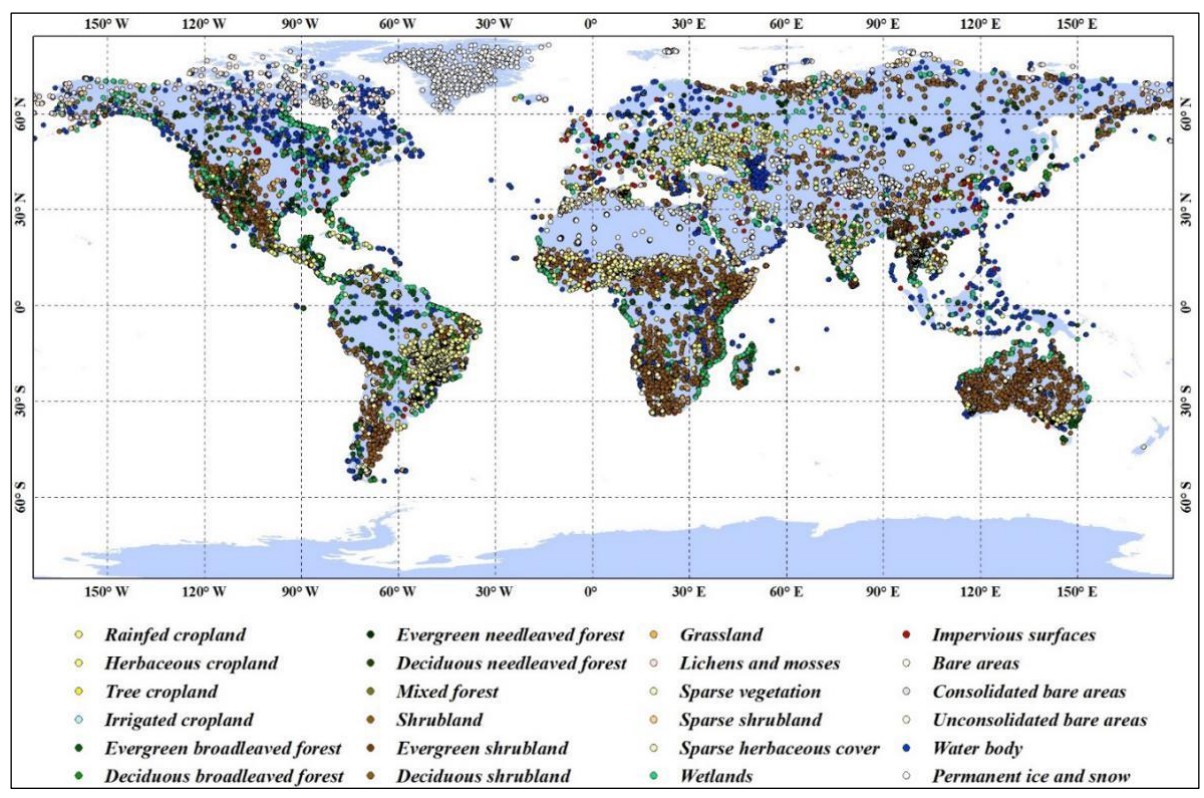

**Figure 2. The spatial distribution of the global validation datasets**

## 3 Methods

### 3.1 Deriving training samples from the GSPECLib

As explained in our previous studies (Zhang et al., 2019; Zhang et al., 2018), the Global Spatial Temporal Spectral Library (GSPECLib) was developed to store the reflectance spectra of different land cover types within each 158.85 km×158.85 km geographic grid cell at a temporal resolution of eight days using time-series of the MCD43A4 NBAR and ESA CCI_LC land-

cover products. The reasons for selecting the CCI_LC and MCD43A4 NBAR products were that: 1) MODIS has similar spectral bands to the Landsat OLI sensor, and MCD43A4 NBAR has better correction for view-angle effects than other SR products such as MOD09A1, meaning that there is more consistency between MCD43A4 NBAR and Landsat 8 SR (at small view angles, i.e. < 15°) (Feng et al., 2012); and 2) the CCI_LC land-cover product has a detailed classification scheme containing 36 land-cover types, achieves the required classification accuracy over homogeneous areas (75.38% overall), and has a relatively high spatial resolution of 300 m as well as a stable transition between the different annual land-cover products (Defourny et al., 2018; Yang et al., 2017b). In contrast to the previous GSPECLib that was used to store the reflectance spectra, the current GSPECLib was developed to derive training samples using the CCI_LC and MCD43A4 NBAR products.

The fine classification system used in this study (Table 2) inherited that of the CCI_LC products after the removal of four mosaic land-cover types (including mosaic natural vegetation and cropland, and mosaic forest and grass or shrubland) because, in the 30-m Landsat imagery, it is possible to clearly identify the mosaic land-cover types in the coarse resolution imagery (Fisher et al., 2018; Mishra et al., 2015). The three wetland land-cover types (tree/shrub/herbaceous cover; flooded; and fresh/saline or brackish water) were further combined into one wetland land-cover type as their high spatial and spectral heterogeneity as well as temporal dynamics made it difficult to identify the wetlands using remote sensing imagery (Gong et al., 2013; Ludwig et al., 2019). It should be noted that the CCI_LC products provide detailed land-cover results only for certain regions and not for the whole world because these detailed land-cover types made use of more accurate and regional information – where available – to define more LCCS classifiers and so to reach a higher level of detail in the legend (Defourny et al., 2018); therefore, the fine classification system in this study simultaneously contained 16 LCCS land-cover types ('multiple-of-ten' values such as 10, 20, 50, 60, …) and 14 detailed regional land-cover types (other 'non-ten' values such as: 11, 12, 61, …).

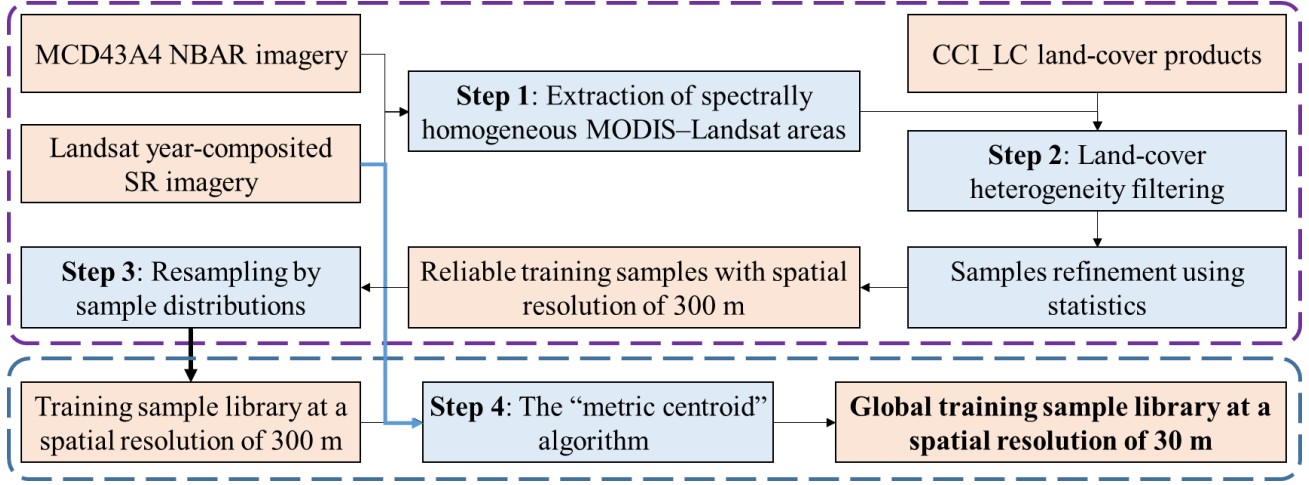

Figure 3. The flowchart of deriving training samples by using multi-source datasets.

Similar to our previous works (Zhang et al., 2019; Zhang et al., 2018), four key steps were adopted to guarantee the confidence of each training point, as illustrated in the Figure 3. As in Zhang et al. (2019), the spectrally homogeneous MODIS–Landsat areas were firstly identified based on the variance of a 3×3 local window using spectral thresholds of [0.03, 0.03, 0.03, 0.06, 0.03, and 0.03] for the six spectral bands (blue, green, red, NIR, SWIR1, and SWIR2) in the both MCD43A4 NBAR products and Landsat SR imagery (Feng et al., 2012). It should be noted that the year-composited Landsat SR data were downloaded from GEE platform with the sinusoidal projection. As the MCD43A4 NBAR is corrected for view-angle effects and Landsat has a small view angle of ±7.5°, the view-angle difference between MCD43A4 and Landsat SR could be considered negligible.

Before the process of refinement and labeling, the CCI_LC land-cover products, which had geographical projections, were reprojected to the sinusoidal projection of MCD43A4. The spatial resolution of MCD43A4 was 1.67 times that of the CCI_LC land-cover product and the spectrally homogeneous MODIS–Landsat areas had been identified in the 3×3 local windows. Also, Defourny et al. (2018) and Yang et al. (2017b) found that the CCI_LC performed better over homogeneous areas; therefore, a larger local 5×5 window was applied to the CCI_LC land-cover product to refine and label each spectrally homogeneous MODIS-Landsat pixel. Specifically, the land-cover heterogeneity in the local 5×5 window was calculated as being the percentages of land-cover types occurring within the window (Jokar Arsanjani et al., 2016a). Aware of the possibility of reprojection and classification errors in the CCI_LC products, the land-cover heterogeneity threshold was empirically selected as approximately 0.95; in other words, if the maximum frequency of dominant land-cover types was less than 22 in the 5×5 window, the point was excluded from GSPECLib. After a spatial–spectral filter had been applied to MCD43A4 and a heterogeneity filter to the CCI_LC product, the points that had homogeneous spectra and land-cover types were retained. In addition, to further remove the abnormal points contaminating by classification error in the CCI_LC, the homogeneous points were refined based on their spectral statistics distribution, in which the normal samples would form the peak of the distribution whereas the influenced samples were on the long tail (Zhang et al., 2018). It should be noted that the geographical coordinates of each homogeneous point were selected as being the center of the local window in the CCI_LC product because this had a higher spatial resolution than that of MCD43A4.

Then, Zhu et al. (2016) and Jin et al. (2014) found that the distribution (proportional to area and equal allocation) and balance of training data had significant impact on classification results, and quantitatively demonstrated that the proportional approach usually achieve higher overall accuracy than the equal allocation distribution. In addition, Zhu et al. (2016) also suggested to extract a minimum of 600 training pixels and a maximum of 8000 training pixels per class for alleviating the problem of unbalancing training data. In this study, the proportional distribution and sample balancing parameters were used to resample these homogeneous points in each GSPECLib 158.85 km×158.85 km geographic grid cell.

**Table 2. The fine classification system and its relationships with other classification systems (LCCS and GlobeLand30 Level 0)**

| Level 0 classification system | LCCS classification system | Id | Fine classification system | Id |
|---|---|---|---|---|
| Cropland | Rain-fed cropland | 10 | Rain-fed cropland | 10 |

| | | | | |
|---|---|---|---|---|
| | | | Herbaceous cover | 11 |
| | | | Tree or shrub cover (Orchard) | 12 |
| | Irrigated cropland | 20 | Irrigated cropland | 20 |
| Forest | Evergreen broadleaved forest | 50 | Evergreen broadleaved forest | 50 |
| | Deciduous broadleaved forest | 60 | Deciduous broadleaved forest | 60 |
| | | | Closed deciduous broadleaved forest | 61 |
| | | | Open deciduous broadleaved forest | 62 |
| | Evergreen needleaved forest | 70 | Evergreen needleaved forest | 70 |
| | | | Closed evergreen needleaved forest | 71 |
| | | | Open evergreen needleaved forest | 72 |
| | Deciduous needleaved forest | 80 | Deciduous needleaved forest | 80 |
| | | | Closed deciduous needleaved forest | 81 |
| | | | Open deciduous needleaved forest | 82 |
| | Mixed-leaf forest | 90 | Mixed-leaf forest | 90 |
| Shrubland | Shrubland | 120 | Shrubland | 120 |
| | | | Evergreen shrubland | 121 |
| | | | Deciduous shrubland | 122 |
| Grassland | Grassland | 130 | Grassland | 130 |
| Wetlands | Wetlands | 180 | Wetlands | 180 |
| Impervious surfaces | Impervious surfaces | 190 | Impervious surfaces | 190 |
| Bare areas | Lichens and mosses | 140 | Lichens and mosses | 140 |
| | Sparse vegetation | 150 | Sparse vegetation | 150 |
| | | | Sparse shrubland | 152 |
| | | | Sparse herbaceous cover | 153 |
| | Bare areas | 200 | Bare areas | 200 |
| | | | Consolidated bare areas | 201 |
| | | | Unconsolidated bare areas | 202 |
| Water body | Water body | 210 | Water body | 210 |
| Permanent ice and snow | Permanent ice and snow | 220 | Permanent ice and snow | 220 |

Lastly, different from the previous spectrally based classification using MCD43A4 reflectance spectra (Zhang et al., 2019), in this study, we proposed to use the Landsat reflectance spectra, derived by combining the global training samples and time-series Landsat imagery, to produce the global 30 m land-cover mapping. However, as the spatial resolution difference between Landsat SR (30 m) and homogeneous training samples (300 m), therefore, the "metric centroid" algorithm proposed by Zhang and Roy (2017) was used to find the optimal and corresponding training points at a resolution of 30 m. Specifically, as each

homogeneous point corresponded to an area equivalent to 10×10 Landsat pixels, the normalized distances (Eq. (2)) between each Landsat pixel and the mean of all 10×10 pixel areas were calculated. The optimal and corresponding training points at 30 m were selected as the ones having the minimum normalized distance,

$$D_i = \left( \rho_i - \frac{1}{n} \sum_{j=1}^{n} \rho_j \right)^2, i = 1,2,\dots,n \tag{2}$$

where $\rho_i$ is a vector representing the annually composited Landsat SR for 2015 and $n$ is the number of Landsat pixels within a 10×10 local window (defined as 100). If several 30-m pixels had the same minimum $D_i$ value then one pixel was selected at random.

## 3.2 Land-cover classification on the GEE platform

Despite the long-term plans for periodic systematic acquisitions and the improved accessibility of Landsat data through global archive consolidation efforts, the availability of Landsat data for persistently cloud-contaminated areas (the rainforest areas in Fig. 1) is less than ideal. To overcome the limitations of scene-level data quality, pixel-based compositing of Landsat data has increased in popularity since the opening of the USGS Landsat archive in 2008 (Griffiths et al., 2013; Woodcock et al., 2008). In particular, the seasonal-composite and metrics-composite are two widely used methods in large-area land-cover classification (Hansen et al., 2014; Massey et al., 2018; Teluguntla et al., 2018; Zhang and Roy, 2017). Recently, Azzari and Lobell (2017) quantitatively demonstrated that season- and metric-based approaches had nearly the same overall accuracies for land-cover classification containing multiple land-cover types or for single cropland mapping. Also, the metrics-composite method proposed by the Hansen et al. (2014) can capture the phenology and land-cover changes without the need for any explicit assumptions or prior knowledge regarding the timing of the season; therefore, its main advantage is that it is applicable globally without the need for location-specific modifications.

In this study, the time-series of Landsat SR imagery and corresponding spectral indexes, including NDVI (Normalized Difference Vegetation index) (Tucker, 1979), NDWI (Normalized Difference Water Index) (Xu, 2006), EVI (Enhanced Vegetation index) (Huete et al., 1999) and NBR (Normalized Burnt Ratio) (Miller and Thode, 2007), were composited into the 25th, 50th and 75th percentiles for each spectral band using the metrics-composite method. It should be noted that the 25th and 75th percentiles were used instead of the minimum and maximum values to minimize the effects of residual haze, cloud and shadows caused by the errors in the CFMask method. In addition, many researchers have found that the texture variables can significantly improve the classification accuracy for land-cover mapping (Li et al., 2017b; Rodriguez-Galiano et al., 2012; Wang et al., 2015; Zhu et al., 2012), for example, Zhu et al. (2012) found that the import of Landsat-derived texture features improved the land-cover accuracy from 86.86% to 92.69%. Therefore, the NIR band texture variables of variance, homogeneity, contrast, dissimilarity, entropy, and correlation were also added using GLCM (Gray Level Co-occurrence Matrix)-based method. Due to the great similarity between the six Landsat optical bands (Rodriguez-Galiano et al., 2012), only the texture variables of the NIR bands were considered. In total, there were 16 spectral–texture metrics ($M_{S-T}$) for each percentile and a

total of 48 metrics for each Landsat pixel. Except for these Landsat-based metrics, the three topographical variables of elevation, slope and aspect, derived from the DEM datasets, were also added.

$$M_{S-T} = \left[ [\rho_b, \rho_g, \rho_r, \rho_{NIR}, \rho_{SWIR1}, \rho_{SWIR2}, NDVI, NDWI, EVI, NBR], [vari, homo, cont, diss, entr, corr]_{NIR} \right] \qquad (3)$$

Afterwards, the random forest (RF) classifier, comprised of a decision-tree classification using the bagging strategy (Breiman, 2001) and an internal algorithm on the GEE platform, was used to combine the training data and aforementioned composited metrics for land-cover mapping. Many studies have demonstrated that the RF performs better with high-dimensional data, gives a higher classification accuracy and is less sensitive to noise and feature selection than other widely used classifiers such

as the support vector machine, artificial neural network, and the classification and regression tree (Belgiu and Drăguţ, 2016; Du et al., 2015; Pelletier et al., 2016). Moreover, the RF classifier has only two adjustable parameters: the number of selected prediction variables (Mtry) and the number of decision trees (Ntree). Belgiu and Drăguţ (2016) also explained that the classification accuracy was less sensitive to Ntree than to the Mtry parameter, and Mtry was usually set to the square root of the number of input variables. due to these advantages, the RF classifier is widely used in land-cover mapping (Gong et al.,

2019; Gong et al., 2013; Zhang and Roy, 2017; Zhang et al., 2019). In this study, the values of Ntree and Mtry were set to 100 and the default value (the square root of the total number of input features), respectively.

There were usually two options for large-area or global land-cover classification including: global classification modelling (Gong et al., 2013; Teluguntla et al., 2018) and local adaptive classification modelling (Gong et al., 2020; Phalke et al., 2020; Zhang et al., 2020). First, the global classification strategy meant using all training samples to train a single classifier which

was suitable for land-cover mapping in any areas. For example, Buchhorn et al. (2020) used 141,000 unique 100×100 m training locations to train a single random forest classifier to generate the Copernicus Global Land Cover layers. Then, the local adaptive classification modelling was firstly divided the globe into a lot of regions and then trained the corresponding local classifiers using the regional training samples, and the global land-cover map was spatially mosaiced by a lot of regional land-cover classification results. For example, Zhang and Roy (2017) split the United States into 561 159×159 km tiles and

then trained 561 corresponding local adaptive random forest models to generate the regional land-cover results, and found the land-cover maps derived from the local adaptive models achieved higher accuracy performance than that of the single global model. Similarly, Radoux et al. (2014) also found that the local adaptive modelling allowed regional tuning of classification parameters to consider regional characteristics and increased the sensitivity of the training samples. Therefore, as illustrated in the previous works, the training samples in a small spatial grid (Landsat scene) might be not enough especially for sparse

land-cover types, and the training samples from neighboring 3 by 3 tiles were also imported (Zhang and Roy, 2017; Zhang et al., 2019), as well as GEE platform had some limitations for computation capacity and memory. Therefore, after balancing the accuracy performance, computation efficiency and training sample volume, the local adaptive random forest models, which split the globe into approximately 948 5°×5° geographical tiles (approximately 3×3 Landsat scenes) similar to our previous work (Zhang et al., 2020), were applied to generate a lot of regional land-cover maps. In addition, to guarantee the spatially

continuous transition over adjacent regional land-cover maps, the training samples from neighboring $3 \times 3$ tiles were used to train the random forest model and classify the central tile.

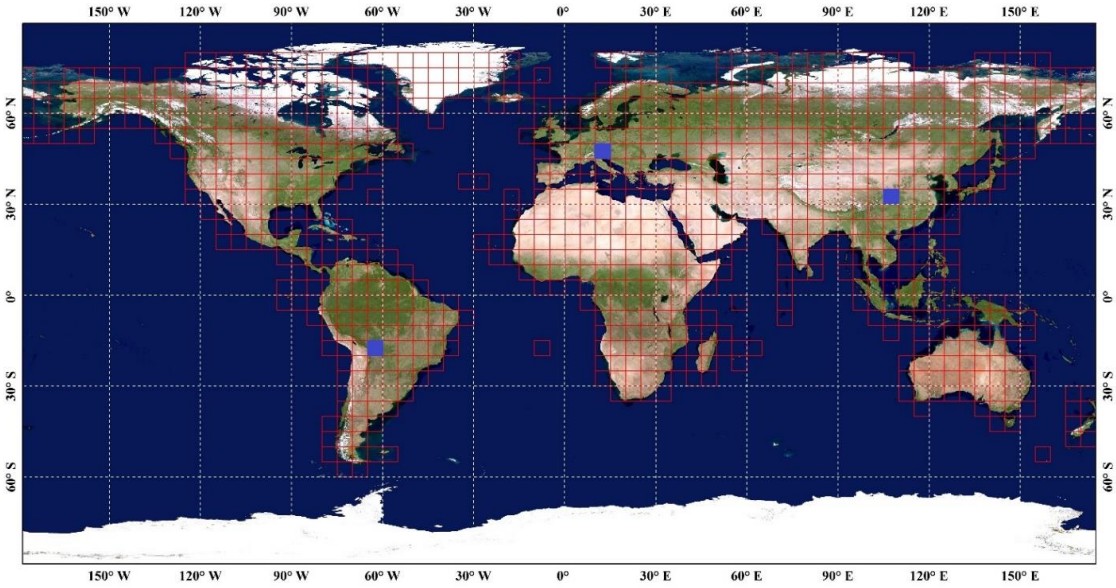

**Figure 4. Overview of the 5°×5°geographical tiles used for local adaptive modelling. Three blue rectangular tiles were used for comparing GLC_FCS30 with other land-cover products. The background imagery came from the National Aeronautics and Space**
**Administration (https://visibleearth.nasa.gov).**

### 3.3 Accuracy assessment

Assessing the accuracy of land-cover products is an essential step in describing the quality of the products before they are used in related applications (Olofsson et al., 2013). In the past, although there has been no standard method of assessing the accuracy of land-cover maps, the error or confusion matrix has been widely considered to be the best measure (Foody and Mathur, 2004;
Gómez et al., 2016; Olofsson et al., 2014). This is because it not only describes the confusion between various land-cover types but also provides quantitative metrics, including the user's accuracy (U.A.) (measuring the commission error), producer's accuracy (P.A.) (measuring the omission error), overall accuracy (O.A.) and kappa coefficient, to measure the performance of the products.

In this study, since the GLC_FCS30 products contained 30 fine land-cover types, including 16 LCCS level-1 types and 14
detailed level-2 types (Table 2), for a more comprehensive validation of the GLC_FCS30 products, the confusion matrices were divided into three parts: 1) a Level-0 confusion matrix containing 9 major land-cover types, similar to the GlobaLand30 and FROM_GLC classification systems; 2) a LCCS Level-1 validation matrix containing 16 level-1 land-cover types, and 3) a LCCS Level-2 validation matrix containing 24 fine land-cover types after the removal of 6 coverage-related level-2 types (closed or open deciduous or evergreen or broadleaved/needle-leaved forests) from the classification system. These 6 coverage-
related types were removed because it was difficult to guarantee the confidence for these detailed land-cover types in the

validation datasets. It should be noted that the relationship between the Level-0 validation system and the classification system used in this study was related to the work of Defourny et al. (2018) and Yang et al. (2017b).

## 4 Results

### 4.1 The GLC_FCS30-2015 land-cover map

Fig. 5 illustrates the global 30-m land-cover map for the nominal year of 2015 (GLC_FCS30-2015) containing 30 fine land-cover types and produced using the time-series of Landsat SR imagery and the local random forest classification models. Intuitively, the GLC_FCS30-2015 land-cover map accurately delineates the spatial distributions of various land-cover types and is consistent with the actual spatial patterns of global land cover: for example, areas of evergreen broadleaved forest are mainly distributed in tropical areas, including the Amazon rainforest, Africa rainforests and India–Malay rainforests, whereas
bare areas are found in the African Sahara, Arabian Desert, Australian deserts and China-Mongolia desert areas. In addition, owing to importing the multi-temporal Landsat features for land-cover classification and using the training samples from neighboring 3 × 3 tiles to train the random forest model and classify the central tile, therefore, the stamping problem that occurs in single-date land-cover classification (Gong et al., 2013; Zhang et al., 2018) has been largely solved in the case of this global map, and the spatial transitions between adjacent geographical tiles are continuous and natural. Similarly, Zhang
and Roy (2017) used the time-series Landsat imagery and imported the neighboring training samples to generate the spatially consistent land-cover classification over the United States.

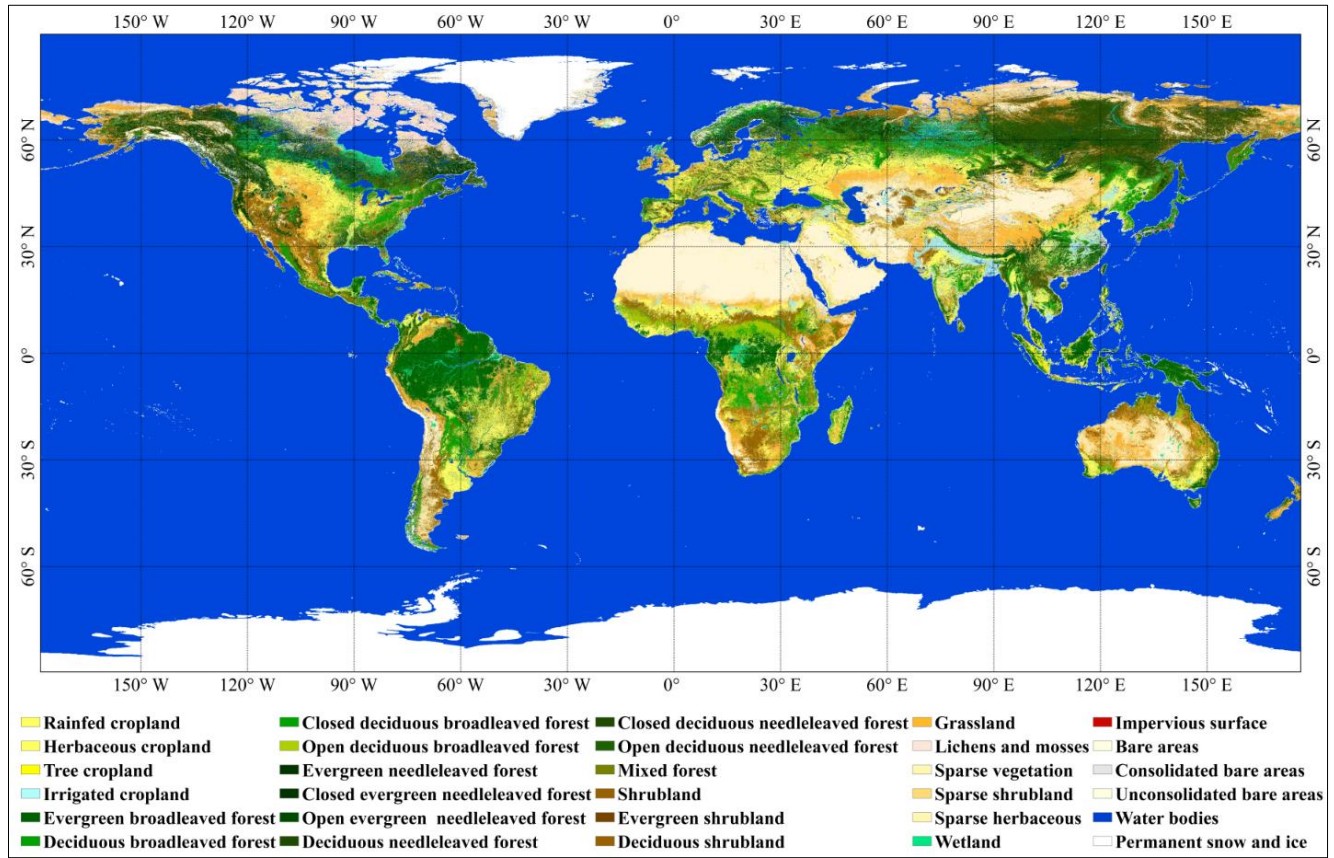

| | | | | |
|---|---|---|---|---|
| ▢ Rainfed cropland | ▢ Closed deciduous broadleaved forest | ▢ Closed deciduous needleleaved forest | ▢ Grassland | ▢ Impervious surface |
| ▢ Herbaceous cropland | ▢ Open deciduous broadleaved forest | ▢ Open deciduous needleleaved forest | ▢ Lichens and mosses | ▢ Bare areas |
| ▢ Tree cropland | ▢ Evergreen needleleaved forest | ▢ Mixed forest | ▢ Sparse vegetation | ▢ Consolidated bare areas |
| ▢ Irrigated cropland | ▢ Closed evergreen needleleaved forest | ▢ Shrubland | ▢ Sparse shrubland | ▢ Unconsolidated bare areas |
| ▢ Evergreen broadleaved forest | ▢ Open evergreen needleleaved forest | ▢ Evergreen shrubland | ▢ Sparse herbaceous | ▢ Water bodies |
| ▢ Deciduous broadleaved forest | ▢ Deciduous needleleaved forest | ▢ Deciduous shrubland | ▢ Wetland | ▢ Permanent snow and ice |

**Figure 5. GLC_FCS30-2015 land-cover map containing 30 fine land-cover types for the nominal year 2015, and the legend colormap inherited from the CCI_LC land-cover product. The legend colormap inherited from the ESA CCI_LC land-cover products**
(Defourny et al., 2018)**.**

Using the validation datasets described earlier, three confusion matrices (Tables 3, 4 & 5) corresponding to different validation systems were generated. Table 3 summarizes the accuracy metrics for 9 major land-cover types: overall, the GLC_FCS30-2015 map achieved an overall accuracy of 82.5% and a kappa coefficient of 0.784. From the perspective of the producer's accuracy, the forest type had the highest accuracy, followed by cropland, permanent ice and snow, bare areas and water body;
wetland, shrubland and grassland had low accuracies. These results indicate that land-cover types that had relatively pure spectral properties or occupied a large proportion of the Earth's surface usually had a relatively high accuracy. In contrast, the complex land-cover types were often confused with other types: for example, the spectra of the wetlands were especially complicated and easily confused with water body and vegetation (Ludwig et al., 2019). As a result, 16.7% and 9.5% wetland validation points were wrongly identified as vegetation (including cropland, forest and shrubland) and water body, respectively,
in Table 3. As for the user's accuracy metric, the accuracy rankings were similar to those for the producer's accuracy; however, in this case, the permanent ice and snow class achieved the highest accuracy.

**Table 3. The accuracy matrix for the GLC_FCS30-2015 land-cover product according to the Level-0 validation scheme and containing 9 major land-cover types**

|  | CRP | FST | SHR | GRS | BaA | WET | IMP | Wat | PIS | Total | P.A. |
|---|---|---|---|---|---|---|---|---|---|---|---|
| **CRP** | 6085 | 338 | 163 | 150 | 70 | 10 | 83 | 18 | 0 | 6917 | 0.880 |
| **FST** | 201 | 12869 | 156 | 54 | 37 | 364 | 2 | 5 | 0 | 13688 | 0.940 |
| **SHR** | 444 | 575 | 3088 | 645 | 576 | 88 | 17 | 6 | 2 | 5441 | 0.568 |
| **GRS** | 197 | 176 | 430 | 3100 | 514 | 171 | 14 | 7 | 0 | 4609 | 0.673 |
| **BaA** | 150 | 109 | 403 | 420 | 7125 | 90 | 3 | 18 | 28 | 8346 | 0.854 |
| **WET** | 78 | 56 | 24 | 23 | 72 | 585 | 15 | 89 | 4 | 946 | 0.618 |
| **IMP** | 52 | 8 | 9 | 12 | 12 | 5 | 384 | 2 | 0 | 484 | 0.793 |
| **Wat** | 48 | 85 | 13 | 7 | 92 | 32 | 3 | 1455 | 1 | 1736 | 0.838 |
| **PIS** | 0 | 16 | 8 | 66 | 89 | 2 | 0 | 47 | 1648 | 1876 | 0.878 |
| **Total** | 7255 | 14232 | 4294 | 4477 | 8587 | 1347 | 521 | 1647 | 1683 | 44043 |  |
| **U.A.** | 0.839 | 0.904 | 0.719 | 0.692 | 0.830 | 0.434 | 0.737 | 0.883 | 0.979 |  |  |
| **O.A.** |  |  |  |  | 0.825 |  |  |  |  |  |  |
| **Kappa** |  |  |  |  | 0.784 |  |  |  |  |  |  |

**Note**: CRP: cropland, FST: forest, SHR: shrubland, GRS: grassland, WET: wetlands, IMP: impervious surfaces, BaA: bare areas, Wat: water body, PIS: permanent ice and snow

Tables 4 & 5 describe the performance of the GLC_FCS30-2015 land-cover map under the LCCS level-1 & 2 validation schemes, respectively. Compared with the values of the accuracy metrics in Table 3, the values in these tables are clearly lower because similar fine land-cover types were easily confused under these conditions. According to Table 4, the GLC_FCS30-2015 achieved an overall accuracy of 71.4% and a kappa coefficient of 0.686. From the perspectives of the user's accuracy and producer's accuracy, there was significant confusion between the forest-related and cropland-related cover types. In order to intuitively display the degree of confusion for the 16 LCCS level-1 land-cover types, the confusion proportions for each of the land-cover types in Table 4 were calculated; these are shown in Fig. 6. First, it can be seen that the complicated land-cover types were more easily misclassified: for example, mixed forest (90) and lichens and mosses (140) had the highest confusion proportions, with more than 60% of the validation samples being misclassified as other types. Secondly, there was a great deal of misclassification between similar land-cover types: for example, more than 20% of irrigated cropland samples (20) were misclassified as rainfed cropland (10), approximately 30% of deciduous needle-leaved forest samples (80) were misclassified as evergreen needle-leaved forest (70), and the confusion between sparse vegetation (150) and bare areas (200) was also considerable.

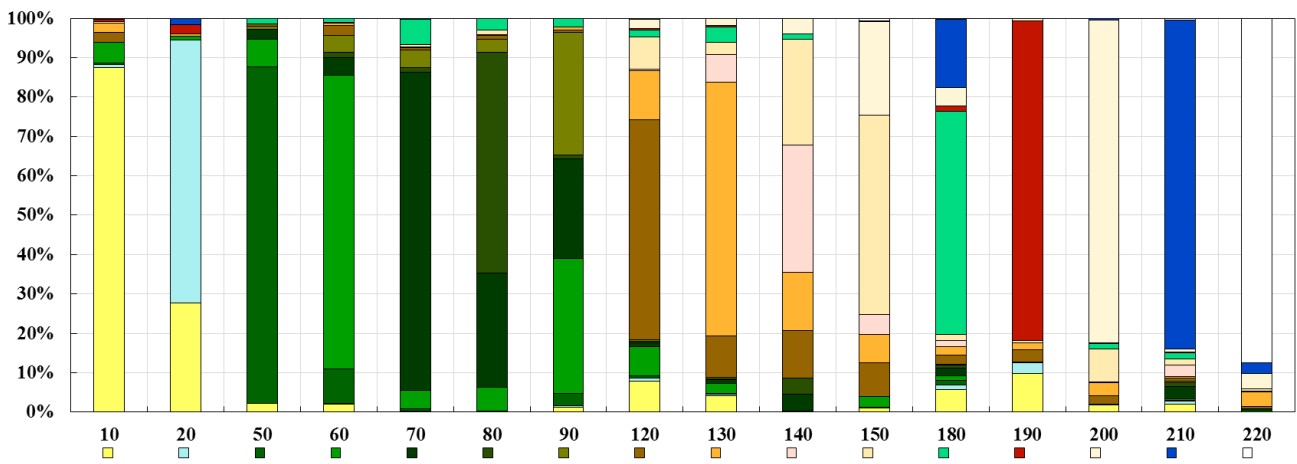

**Figure 6. The confusion proportions for each of the land-cover types in the LCCS level-1 validation scheme.**

In the Table 5, it can be seen that GLC_FCS30-2015 achieved an overall accuracy of 68.7% and kappa coefficient of 0.662. It should be noted that the yellow marks in Table 5 also represented they were correctly classified because GLC_FCS30-2015 simultaneously consisted of 16 LCCS land-cover types (the 'tens' values such as 10, 20, 50 etc.) and 14 detailed regional land-cover types (the 'non-ten' values such as: 11, 12, 61 etc.) which were only present in some regions (Defourny et al., 2018).

Also, the 14 detailed land-cover types simultaneously belonged to the corresponding LCCS land-cover types according to the Table 2; similar operators for these detailed land-cover types can also be found in the works of Defourny et al. (2018) (see Table 3-7) and Bontemps et al. (2010). Under the LCCS level-2 fine validation system, the accuracy metrics were basically consistent with those found for the LCCS level-1 validation scheme. Fig. 7 illustrates the confusion proportions between each of the fine land-cover types. In contrast to the results discussed above, the degrees of confusion for these fine land-cover types

is more significant: for example, most tree-covered cropland (12) samples are misclassified as herbaceous-covered cropland (11), and the confusion between the LCCS land-cover types (the 'tens' values) and the corresponding detailed land-cover types (the 'non-ten' values) is more obvious.

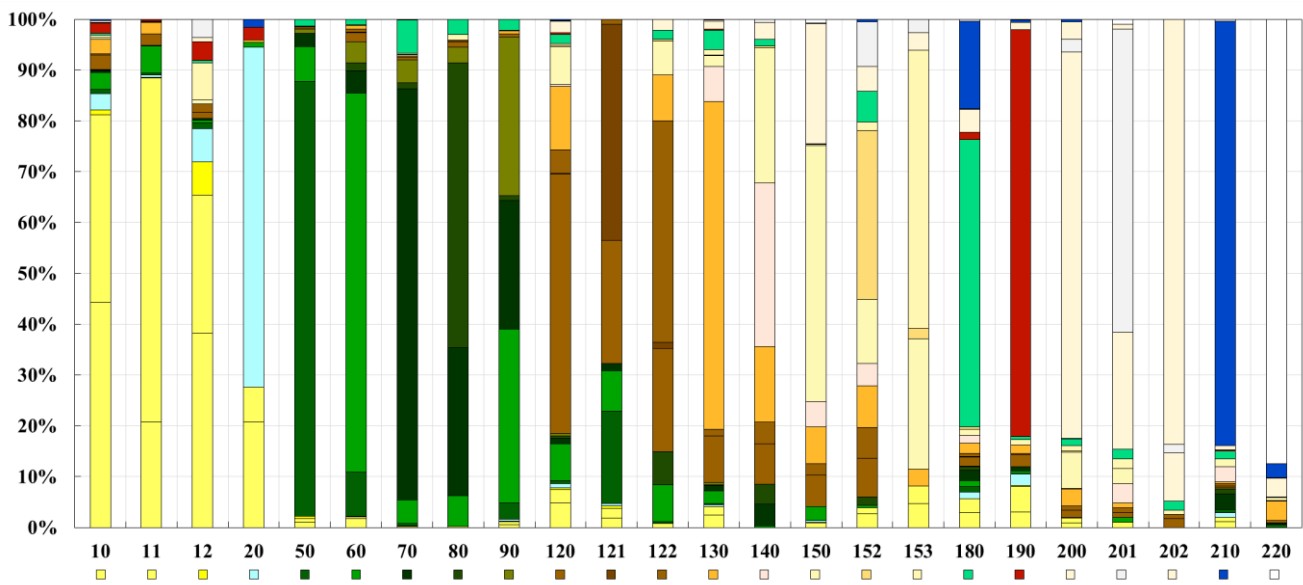

**Figure 7. The confusion proportions for each of the land-cover types in LCCS level-2 validation scheme.**

**Table 4. The accuracy matrix for the GLC_FCS30-2015 land-cover product according to the LCCS level-1 validation scheme.**

|  | 10 | 20 | 50 | 60 | 70 | 80 | 90 | 120 | 130 | 140 | 150 | 180 | 190 | 200 | 210 | 220 | Total | P. A. |
|---|---|---|---|---|---|---|---|---|---|---|---|---|---|---|---|---|---|---|
| 10 | 5305 | 86 | 32 | 281 | 12 | 1 | 4 | 163 | 146 | 0 | 47 | 10 | 66 | 23 | 5 | 0 | 6181 | 0.858 |
| 20 | 213 | 481 | 0 | 8 | 0 | 0 | 0 | 0 | 4 | 0 | 0 | 0 | 17 | 0 | 13 | 0 | 736 | 0.654 |
| 50 | 65 | 0 | 2830 | 152 | 82 | 0 | 28 | 17 | 1 | 0 | 0 | 47 | 0 | 0 | 0 | 0 | 3222 | 0.878 |
| 60 | 82 | 3 | 325 | 3010 | 175 | 58 | 189 | 99 | 28 | 0 | 10 | 44 | 1 | 1 | 2 | 0 | 4027 | 0.747 |
| 70 | 10 | 0 | 12 | 136 | 2469 | 34 | 133 | 15 | 7 | 1 | 10 | 192 | 1 | 2 | 3 | 0 | 3025 | 0.816 |
| 80 | 2 | 0 | 0 | 59 | 283 | 545 | 31 | 11 | 2 | 0 | 11 | 29 | 0 | 0 | 0 | 0 | 973 | 0.560 |
| 90 | 31 | 8 | 67 | 840 | 604 | 24 | 783 | 14 | 16 | 0 | 1 | 52 | 0 | 1 | 0 | 0 | 2441 | 0.321 |
| 120 | 402 | 42 | 64 | 395 | 57 | 39 | 20 | 3088 | 645 | 21 | 422 | 88 | 17 | 133 | 6 | 2 | 5441 | 0.568 |
| 130 | 183 | 14 | 9 | 94 | 47 | 7 | 19 | 430 | 3100 | 311 | 128 | 171 | 14 | 75 | 7 | 0 | 4609 | 0.673 |
| 140 | 0 | 0 | 0 | 1 | 13 | 12 | 0 | 35 | 39 | 93 | 83 | 5 | 0 | 12 | 0 | 0 | 293 | 0.317 |
| 150 | 47 | 8 | 0 | 75 | 0 | 3 | 0 | 254 | 218 | 147 | 1540 | 13 | 0 | 692 | 4 | 26 | 3027 | 0.509 |
| 180 | 64 | 14 | 12 | 12 | 22 | 8 | 2 | 24 | 23 | 12 | 17 | 585 | 15 | 43 | 89 | 4 | 946 | 0.618 |
| 190 | 38 | 14 | 1 | 1 | 4 | 1 | 1 | 9 | 12 | 0 | 5 | 5 | 384 | 7 | 2 | 0 | 484 | 0.793 |
| 200 | 94 | 1 | 0 | 2 | 3 | 0 | 0 | 114 | 163 | 14 | 415 | 72 | 3 | 4129 | 14 | 2 | 5026 | 0.822 |
| 210 | 33 | 15 | 3 | 4 | 57 | 17 | 4 | 13 | 7 | 49 | 28 | 32 | 3 | 15 | 1455 | 1 | 1736 | 0.838 |
| 220 | 0 | 0 | 2 | 6 | 6 | 0 | 2 | 8 | 66 | 2 | 13 | 2 | 0 | 74 | 47 | 1648 | 1876 | 0.878 |
| Total | 6569 | 686 | 3357 | 5076 | 3834 | 749 | 1216 | 4294 | 4477 | 650 | 2730 | 1347 | 521 | 5207 | 1647 | 1683 | 44043 | |
| U. A. | 0.808 | 0.701 | 0.843 | 0.593 | 0.644 | 0.728 | 0.644 | 0.719 | 0.692 | 0.143 | 0.564 | 0.434 | 0.737 | 0.793 | 0.883 | 0.979 | | |

O. A.  0.714

Kappa  0.686

**Table 5. The accuracy matrix for the GLC_FCS30-2015 land-cover product according to the LCCS level-2 validation scheme.**

| | 10 | 11 | 12 | 20 | 50 | 60 | 70 | 80 | 90 | 120 | 121 | 122 | 130 | 140 | 150 | 152 | 153 | 180 | 190 | 200 | 201 | 202 | 210 | 220 | Total | P.A. |
|---|---|---|---|---|---|---|---|---|---|---|---|---|---|---|---|---|---|---|---|---|---|---|---|---|---|---|
| 10 | 902 | 747 | 19 | 54 | 15 | 68 | 9 | 1 | 4 | 58 | 0 | 5 | 61 | 0 | 7 | 0 | 11 | 9 | 40 | 3 | 8 | 0 | 5 | 0 | 2026 | 0.823 |
| 11 | 823 | 2654 | 5 | 17 | 14 | 212 | 2 | 0 | 0 | 93 | 0 | 0 | 85 | 0 | 5 | 0 | 4 | 0 | 17 | 1 | 0 | 0 | 0 | 0 | 3932 | 0.884 |
| 12 | 91 | 48 | 16 | 15 | 3 | 1 | 1 | 0 | 0 | 3 | 0 | 4 | 0 | 0 | 2 | 0 | 18 | 1 | 9 | 2 | 9 | 0 | 0 | 0 | 223 | 0.480 |
| 20 | 160 | 53 | 0 | 481 | 0 | 8 | 0 | 0 | 0 | 0 | 0 | 0 | 4 | 0 | 0 | 0 | 0 | 0 | 17 | 0 | 0 | 0 | 13 | 0 | 736 | 0.654 |
| 50 | 29 | 22 | 14 | 0 | 2830 | 152 | 82 | 0 | 28 | 1 | 15 | 1 | 1 | 0 | 0 | 0 | 0 | 47 | 0 | 0 | 0 | 0 | 0 | 0 | 3222 | 0.878 |
| 60 | 67 | 14 | 1 | 3 | 325 | 3010 | 175 | 58 | 189 | 71 | 3 | 25 | 28 | 0 | 10 | 0 | 0 | 44 | 1 | 0 | 1 | 0 | 2 | 0 | 4027 | 0.747 |
| 70 | 4 | 6 | 0 | 0 | 12 | 136 | 2469 | 34 | 133 | 14 | 0 | 1 | 7 | 1 | 7 | 3 | 0 | 192 | 1 | 2 | 0 | 0 | 3 | 0 | 3025 | 0.816 |
| 80 | 0 | 2 | 0 | 0 | 0 | 59 | 283 | 545 | 31 | 10 | 1 | 0 | 2 | 0 | 11 | 0 | 0 | 29 | 0 | 0 | 0 | 0 | 0 | 0 | 973 | 0.560 |
| 90 | 14 | 14 | 3 | 8 | 67 | 840 | 604 | 24 | 783 | 14 | 0 | 0 | 16 | 0 | 1 | 0 | 0 | 52 | 0 | 1 | 0 | 0 | 0 | 0 | 2441 | 0.321 |
| 120 | 240 | 131 | 21 | 41 | 28 | 359 | 54 | 22 | 20 | 2526 | 4 | 242 | 623 | 21 | 370 | 12 | 21 | 83 | 17 | 115 | 10 | 2 | 6 | 2 | 4970 | 0.558 |
| 121 | 4 | 3 | 1 | 1 | 35 | 17 | 3 | 0 | 0 | 50 | 91 | 2 | 0 | 0 | 0 | 0 | 0 | 0 | 0 | 0 | 0 | 0 | 0 | 0 | 207 | 0.681 |
| 122 | 0 | 2 | 0 | 0 | 1 | 19 | 0 | 17 | 0 | 51 | 3 | 119 | 22 | 0 | 18 | 1 | 0 | 5 | 0 | 6 | 0 | 0 | 0 | 0 | 264 | 0.644 |
| 130 | 106 | 77 | 0 | 14 | 9 | 94 | 47 | 7 | 19 | 374 | 0 | 56 | 3100 | 311 | 94 | 5 | 29 | 171 | 14 | 65 | 1 | 9 | 7 | 0 | 4609 | 0.673 |
| 140 | 0 | 0 | 0 | 0 | 0 | 1 | 13 | 12 | 0 | 24 | 0 | 11 | 39 | 93 | 82 | 1 | 0 | 5 | 0 | 10 | 2 | 0 | 0 | 0 | 293 | 0.317 |
| 150 | 25 | 3 | 0 | 8 | 0 | 74 | 0 | 0 | 0 | 170 | 0 | 61 | 198 | 139 | 1325 | 0 | 8 | 2 | 0 | 653 | 0 | 1 | 3 | 26 | 2696 | 0.491 |
| 152 | 5 | 2 | 0 | 0 | 0 | 1 | 0 | 3 | 0 | 12 | 0 | 11 | 15 | 8 | 22 | 60 | 3 | 11 | 0 | 13 | 16 | 0 | 1 | 0 | 183 | 0.328 |
| 153 | 7 | 5 | 0 | 0 | 0 | 0 | 0 | 0 | 0 | 0 | 0 | 0 | 5 | 0 | 38 | 3 | 81 | 0 | 0 | 5 | 4 | 0 | 0 | 0 | 148 | 0.547 |
| 180 | 31 | 33 | 0 | 14 | 12 | 12 | 22 | 8 | 2 | 18 | 1 | 5 | 23 | 12 | 13 | 4 | 0 | 585 | 15 | 42 | 1 | 0 | 89 | 4 | 946 | 0.618 |
| 190 | 15 | 21 | 2 | 14 | 1 | 1 | 4 | 1 | 1 | 8 | 0 | 1 | 12 | 0 | 4 | 0 | 1 | 5 | 384 | 6 | 1 | 0 | 2 | 0 | 484 | 0.793 |
| 200 | 42 | 49 | 2 | 1 | 0 | 1 | 3 | 0 | 0 | 69 | 0 | 40 | 162 | 10 | 345 | 12 | 52 | 68 | 3 | 3643 | 122 | 167 | 14 | 1 | 4806 | 0.818 |
| 201 | 1 | 0 | 0 | 0 | 0 | 1 | 0 | 0 | 0 | 1 | 0 | 1 | 1 | 4 | 3 | 0 | 2 | 2 | 0 | 24 | 62 | 1 | 0 | 1 | 104 | 0.827 |
| 202 | 0 | 0 | 0 | 0 | 0 | 0 | 0 | 0 | 0 | 2 | 0 | 1 | 0 | 0 | 1 | 0 | 0 | 2 | 0 | 11 | 2 | 97 | 0 | 0 | 116 | 0.931 |
| 210 | 18 | 15 | 0 | 15 | 3 | 4 | 57 | 17 | 4 | 7 | 0 | 6 | 7 | 49 | 28 | 0 | 0 | 32 | 3 | 15 | 0 | 0 | 1455 | 1 | 1736 | 0.838 |
| 220 | 0 | 0 | 0 | 0 | 2 | 6 | 6 | 0 | 2 | 8 | 0 | 0 | 66 | 2 | 13 | 0 | 0 | 2 | 0 | 72 | 2 | 0 | 47 | 1648 | 1876 | 0.878 |
| Total | 2584 | 3901 | 84 | 686 | 3357 | 5076 | 3834 | 749 | 1216 | 3584 | 118 | 592 | 4477 | 650 | 2399 | 101 | 230 | 1347 | 521 | 4689 | 241 | 277 | 1647 | 1683 | 44043 | |
| U.A. | 0.703 | 0.872 | 0.417 | 0.701 | 0.843 | 0.593 | 0.644 | 0.728 | 0.644 | 0.733 | 0.805 | 0.610 | 0.692 | 0.143 | 0.577 | 0.594 | 0.387 | 0.434 | 0.737 | 0.784 | 0.763 | 0.953 | 0.883 | 0.979 | | |
| O.A. | | | | | | | | | | | | 0.687 | | | | | | | | | | | | | | |
| Kappa | | | | | | | | | | | | 0.662 | | | | | | | | | | | | | | |

**4.2 Comparison between GLC_FCS30-2015 and other land-cover products**

**4.2.1 Comparison between three global 30 m land-cover products**

Based on the global validation datasets and the Level-0 validation scheme, the classification accuracy of GLC_FCS30-2015 was compared to other two global 30-m land-cover products (FROM_GLC-2015 and GlobeLand30-2010), as listed in the Table 6. Overall, the GLC_FCS30-2015 achieved the best accuracy performance of 82.5% against the FROM_GLC-2015 of 59.1% and the GlobeLand30-2010 of 75.9%. Specifically, the GLC_FCS30-2015 gave better performance than GlobeLand30-2010 in shrublands, grasslands and impervious surfaces, and achieved similar accuracies with the GlobeLand30 in most land-cover types (cropland, forest, bare land, water body and permanent ice and snow). Compared to the FROM_GLC-2015 products, the GLC_FCS30-2015 and GlobeLand30-2010 had higher accuracy for most land-cover types especially for the cropland and forest.

**Table 6. The accuracy metrics of three global 30 m land-cover products using the validation datasets.**

| | | CRP | FST | SHR | GRS | BaA | WET | IMP | Wat | PIS | O.A. | Kappa |
|---|---|---|---|---|---|---|---|---|---|---|---|---|
| GLC_FCS30-2015 | P.A. | 0.880 | 0.940 | 0.568 | 0.673 | 0.854 | 0.618 | 0.793 | 0.838 | 0.878 | 0.825 | 0.784 |
| | U.A. | 0.839 | 0.904 | 0.719 | 0.692 | 0.830 | 0.434 | 0.737 | 0.883 | 0.979 | | |
| FROM_GLC-2015 | P.A. | 0.477 | 0.749 | 0.294 | 0.484 | 0.696 | 0.033 | 0.459 | 0.781 | 0.647 | 0.591 | 0.499 |
| | U.A. | 0.747 | 0.771 | 0.500 | 0.263 | 0.638 | 0.484 | 0.771 | 0.346 | 0.962 | | |
| GlobeLand30-2010 | P.A. | 0.882 | 0.926 | 0.323 | 0.586 | 0.725 | 0.526 | 0.814 | 0.891 | 0.908 | 0.759 | 0.704 |
| | U.A. | 0.887 | 0.905 | 0.617 | 0.367 | 0.776 | 0.384 | 0.889 | 0.908 | 0.992 | | |

**Note**: CRP: cropland, FST: forest, SHR: shrubland, GRS: grassland, WET: wetlands, IMP: impervious surfaces, BaA: bare areas, Wat: water body, PIS: permanent ice and snow

Similarly, Kang et al. (2020) also analysed the performance of three global land-cover products in the complicated tropical rainforest region (Indonesia) using exceeding 2000 verification points, and validation results indicated that the GLC_FCS-2015 achieved the highest accuracy of 65.59%, followed by the GlobeLand30-2010 of 61.65% and FROM_GLC-2015 of 57.71%, specifically, all the three land-cover products had greater performance for forests and impervious surfaces, and the cropland and wetland mapping accuracy of GLC_FCS30-2015 were higher than that of the other two products (Kang et al., 2020).

Except for the quantitative assessment, three 5°×5° typical regions (the blue rectangles in Fig. 4) and their local enlargements, covering various climate and landscape environment, were selected to directly illustrate the performance of each land-cover product in Fig. 8. Overall, there was higher spatial consistency between the GLC_FCS30-2015 and GlobeLand30-2010 products, both of them accurately depicted the spatial distributions of different land-cover types. As for the FROM_GLC-2015 products, it was different from other two products in spatial distribution, for example, the areas (in the Figure 8II), identified by FROM_GLC-2015 as grassland and shrubland, were labelled as cropland and forest in the GLC_FCS30-2015 and GlobeLand30-2010. In addition, from the perspective of land-cover diversity, it was obvious that the GLC_FCS30-2015 products had significant advantages over other two products which made the regional land-cover maps of GLC_FCS30-2015

contain diverse colour legends. In more detail, as for the cropland-prevalent areas (Fig. 8I-a and III-c), the spatial distribution of GLC_FCS30-2015 was similar to the GlobeLand30-2010 products, however, the FROM_GLC-2015 had omission error for impervious surfaces (Fig. 8I-a) and misidentified some cropland pixels as grassland (Fig. 8I-a) and forest (Fig. 8III-c). Secondly, for the undulating agricultural and forestry areas (Fig. 8I-b, 8I-c, 8II-b, 8III-a), three land-cover products captured

the spatial patterns of various land-cover types, for example, the cropland usually located in the flat areas, and the mountain areas mainly contained the forest and grassland.  Lastly, in the woodland areas where some forests are reclaimed as farmland (Fig. 8II-a), both the GLC_FCS30-2015 and GlobeLand30-2010 accurately delineated the tracks of human interference, and the GLC_FCS30-2015 had larger cropland areas than that of GlobeLand30-2010 which also demonstrated the increasing of reclamation over the 5-years interval. Different from other two products, the FROM_GLC-2015 identified these reclaimed

areas as the grassland pixels and some forest pixels also labelled as grassland which made the FROM_GLC-2015 had largest grassland area in the Fig. 8-IIa.

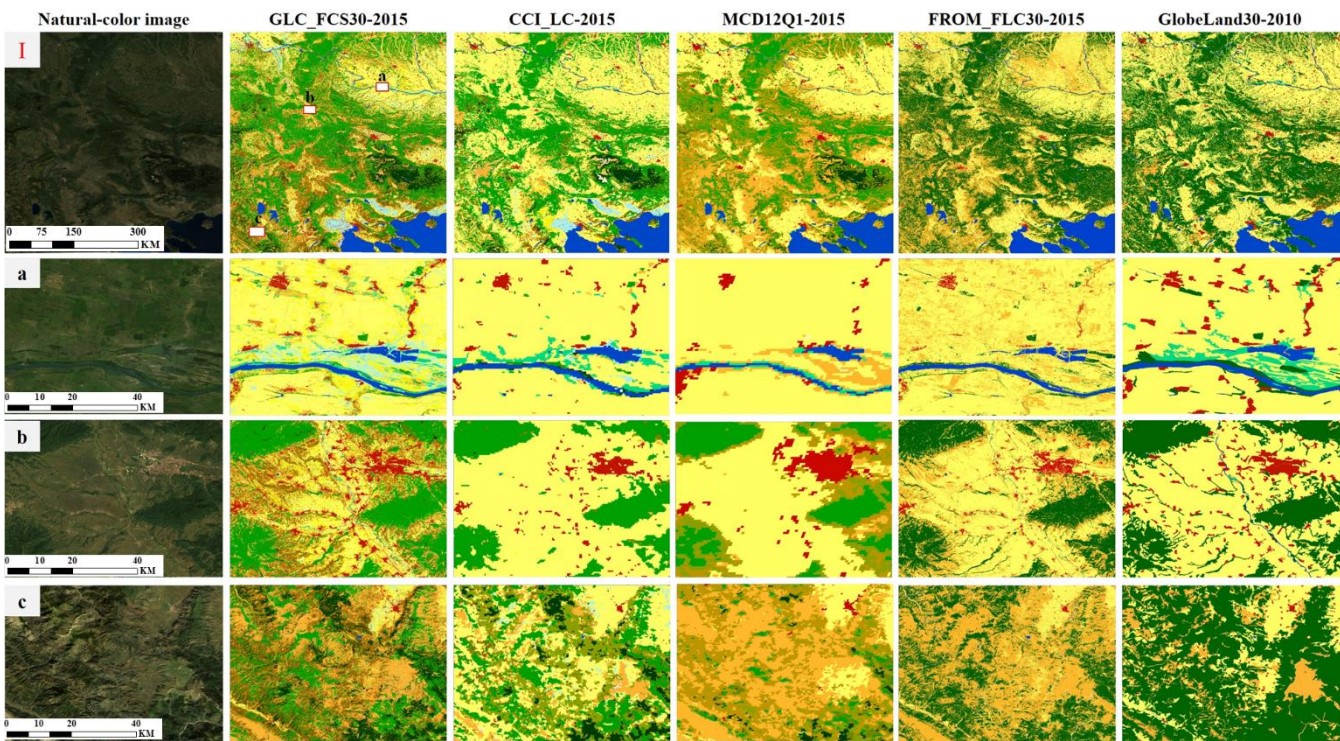

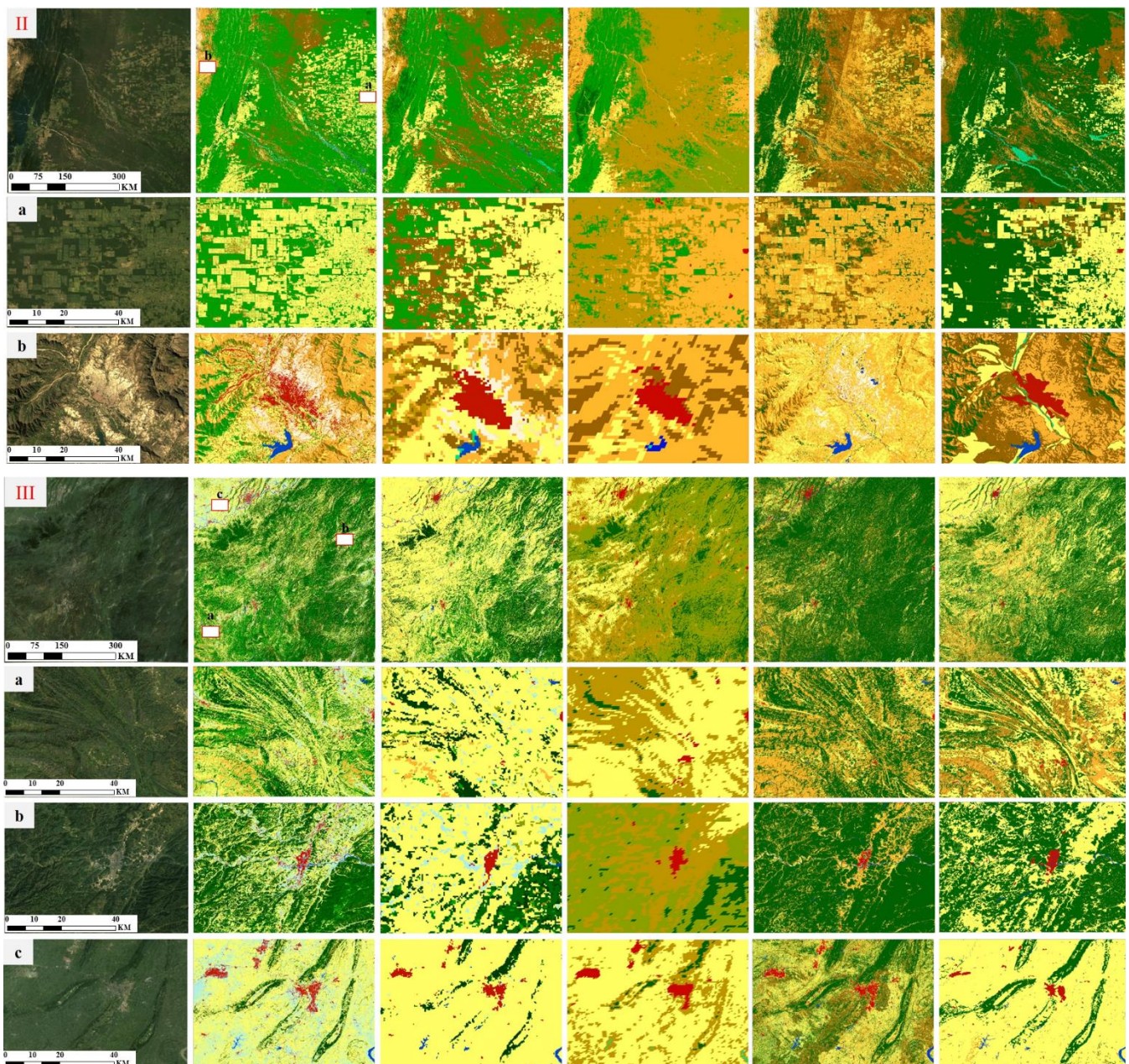

Figure 8. Comparison between GLC_FCS30-2015 and other land-cover products (CCI_LC-2015 products developed by (Defourny et al., 2018), the MCD12Q1-2015 developed by (Friedl et al., 2010), the FROM_GLC-2015 developed by (Gong et al., 2013) and the GlobeLand30 developed by (Chen et al., 2015)) in three 5°×5° regions. In each case, 2–3 local enlargements (a-c) with the size of 40km×60 km were used to reveal further details of each land-cover product.

**4.2.2 The comparisons between GLC_FCS30-2015 with CCI_LC and MCD12Q1 land-cover products**

Except for comparing with global 30-m land-cover products, two widely used global products (CCI_LC-2015 and MCD12Q1), which both contained diverse land-cover types, were also selected to comprehensively analyse performance of the GLC_FCS30-2015. It should be noted that the global validation dataset (section 2.3) was collected to validate the 30-m land-cover products, so the quantitative assessment was skipped for the coarse resolution land-cover products of CCI_LC-2015 and MCD12Q1-2015. Fig. 8 intuitively compared the performances of GLC_FCS30-2015, CCI_LC-2015 and MCD12Q1-2015 products over three 5°×5° typical regions and corresponding local enlargements. Overall, the spatial consistency of GLC_FCS30-2015 and CCI_LC-2015 was higher than that of MCD12Q1-2015 because the GLC_FCS30-2015 and CCI_LC-2015 shared same classification system. For example, the savannas pixels (tree cover 10%-30%) (Friedl et al., 2010) in the MCD12Q1-2015 were labelled as broadleaved forest in the other two products (Fig. 8II).

Lastly, it can be found that the GLC_FCS30-2015 had a great advantage in spatial details compared to the CCI_LC-2015 and MCD12Q1-2015 products over these local enlargements in Fig. 8. For example, the river boundary in the Fig. 8I-a, the fragmented impervious surfaces in the Fig. 8I-a, 8I-b and 8II-b, and the terrain changes in Fig. 8I-c, II-b and III-a, were more accurately captured in the GLC_FCS30-2015, while two coarse land-cover products (CCI_LC-2015 and MCD12Q1-2015) usually lost these details. Therefore, compared with CCI_LC-2015 and MCD12Q1-2015 land-cover products, the GLC_FCS30-2015 not only had obvious advantages in spatial details, but also achieved a higher accuracy and corrected a lot of misclassification in the CCI_LC-2015 land-cover products.

## 5 Discussion

### 5.1 Advantages of GLC_FCS30 using huge training samples

Global land-cover classification is a challenging and labor-intensive task because of the large-volume of data pre-processing involved, the high-performance computing requirements, and the difficulty of collecting training data that allows the classification models to be both locally reliable and globally consistent (Friedl et al., 2010; Giri et al., 2013; Zhang and Roy, 2017). Thanks to the parallel computing ability and efficient and free access to multi-petabyte, analysis-ready remote-sensing data that is available on the GEE platform (Gorelick et al., 2017), the main challenge lies in collecting sufficient reliable training data. In this study, we proposed to extend our previous work on SPECLib-based classifications (Zhang et al., 2019; Zhang et al., 2018) and to derive global high-quality training data from the updated GSPECLib for global land-cover mapping (Section 3.1). Figure 9 illustrates the number of global training samples in each 1°×1° geographical grid cell. The statistics are generally consistent with the land-cover patterns shown in Fig. 5. In addition, in contrast to other studies that used manual interpretation of samples for global land-cover mapping (Friedl et al., 2010; Gong et al., 2013; Tateishi et al., 2014), the total number of training samples in this study reaching 27,858,258 points and so was tens to hundreds of times higher than that used in these global land-cover classifications.

To demonstrate the importance of sample sizes, 200,000 points, approximately 1% of total training samples, were randomly selected to quantitatively analyse the relationship between overall accuracy and the corresponding sample size. Specifically, we used the 10-fold cross-validation method to split these points into training and validation samples, and then gradually increase the size of training samples with the step of 2% and repeat the process for 100 times. Figure 10a illustrated the overall accuracy (Level-0 and LCCS level-1 classification systems) increased for the increased percentage of training samples. It was

found that the overall accuracy rapidly increased when the percentage of training samples increased from 1% to 30%, while it remained relatively stable when the percentage of training samples was higher than 30%. Therefore, the appropriate sample size should be larger than the 60,000 (30% of the total input points), fortunately, the local training samples in this study almost all exceeded the 60,000 because the training samples from neighboring 3 × 3 tiles were used to train the random forest model and classify the central tile. Similarly, Foody (2009) also found that the sample size had a positive relationship with the

classification accuracy up to the point where the sample size was saturated, and Zhu et al. (2016) suggested that the optimal size was a total of 20,000 training pixels to classify an area about the size of a Landsat scene.

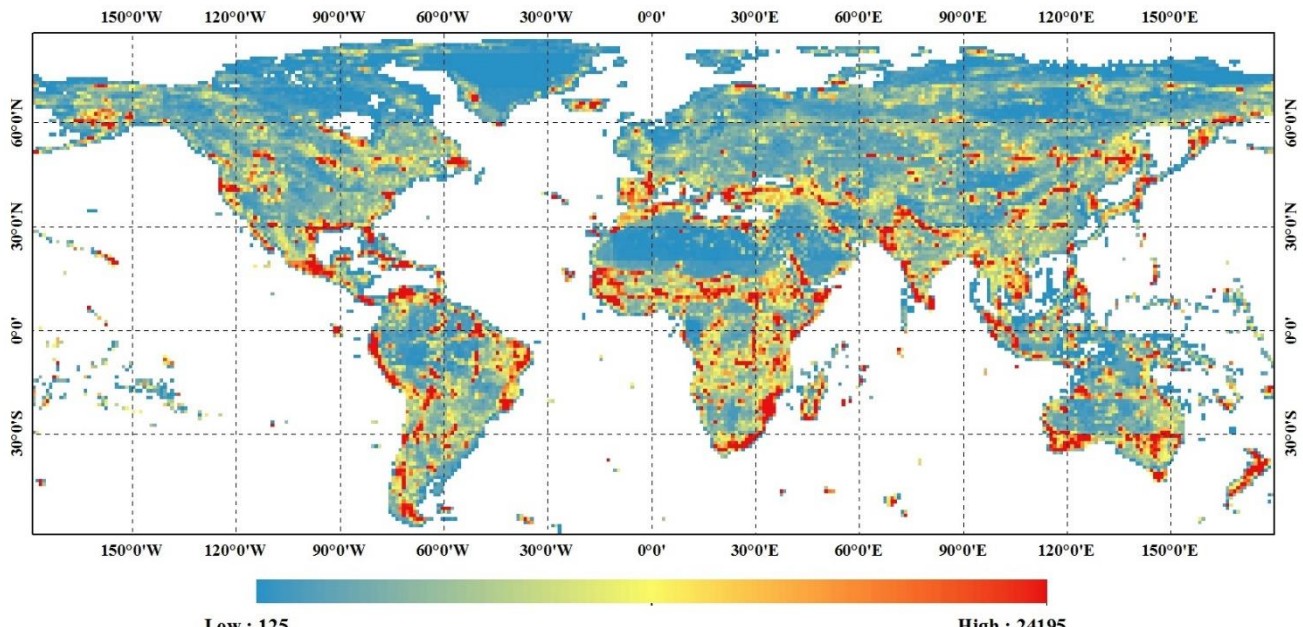

**Figure 9. The number of global training samples in each 1°×1° geographical grid cell.**

Secondly, many studies have demonstrated that the sample outliers had influence on the land-cover classification accuracy

(Mellor et al. 2015, Pelletier et al. 2017). In this study, using previous 200,000 training points, we further analyzed the relationship between overall classification accuracy and erroneous training sample by randomly changing the category of a certain percentage of these samples and using the "noisy" samples to train the random forest classifier. Similar to the previous quantitative analysis of sample size, we gradually increased the percentage of erroneous training samples with the step of 2% and then repeat the process for 100 times. Figure 10b showed that the overall accuracy of two classification systems (level-0

and LCCS level-1) generally decreased with the increasing of percentage of erroneous sample points. It remained relatively stable when the percentage of erroneous training sample was controlled within 30%, and decreased obviously after exceeding the threshold of 30%. Meanwhile, the overall accuracy of simple classification system was more susceptible to the erroneous samples than that of the LCCS classification system in the Figure 10b. Similarly, many scientists have also demonstrated that a small number of erroneous training data have little effect on the classification results (Gong et al., 2019; Mellor et al., 2015;

Pelletier et al., 2016; Zhu et al., 2016): for example, Mellor et al. (2015) found the error rate of the RF classifier was insensitive to mislabeled training data, and the overall accuracy decreased from 78.3% to 70.1% when the proportion of mislabeled training data increased from 0% to 25%. Similarly, Pelletier et al. (2016) found the RF classifier was little affected by low random noise levels up to 25%−30% but that the performance dropped at higher noise levels.

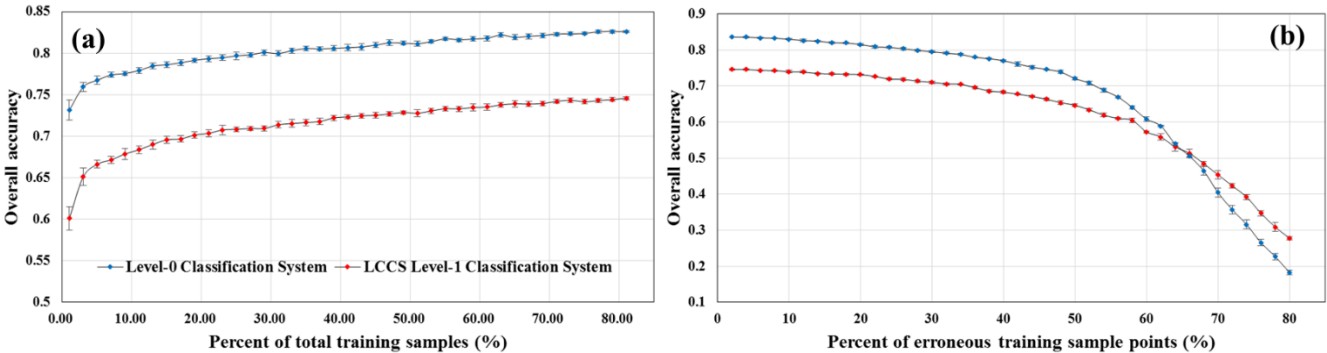

**Figure 10. Sensitivity analysis showing the relations between the overall classification accuracy and the percentage of total samples and erroneous sample points.**

Defourny et al. (2018) demonstrated that CCI_LC achieved an overall accuracy of 75.38% for homogeneous areas. In this study, some measures have been taken to guarantee the confidence of training samples. Some complicated land-cover types were then further optimized to improve the accuracy of the training data; for example, impervious surfaces were imported as

an independent product and directly superimposed over the final global land-cover classifications, the three wetland types were merged into an overall wetland land-cover type, and four mosaicked land-cover types were removed (Table 2). After optimizing these complicated land-cover types, the overall accuracy of CCI_LC reached 77.36% for homogeneous areas based on the confusion matrix of Defourny et al. (2018). In addition, other measures, including the spectral filters applied to the MCD43A4 NBAR data, the land-cover homogeneity constraint for CCI_LC land-cover products, and the "metric centroid"

algorithm for removing the resolution differences, were used to further improve confidence in the training data. Therefore, a part of training samples (exceeding 18000 points) in the previous analysis were randomly selected to quantitatively evaluate the confidence of the global training dataset, after pixel-by pixel interpretation and inspection, the validation results indicated that these samples had satisfactory performance with the overall accuracy of 91.7% for the Level-0 classification system and 82.6% for Level-1 LCCS classification system. Therefore, it can be assumed that the training data, derived by combining the

MCD43A4 NBAR and CCI_LC land-cover products, were accurate and suitable for large-area land-cover mapping at 30 m.

Lastly, the sample balance is also an important factor in land-cover classification especially for rare land-cover types, because unbalanced training data would cause the under-fitting of classification model for rare land-cover types and further degrade the classification accuracy. In this study, we used the sample balancing parameters (a minimum of 600 training pixels and a maximum of 8000 training pixels per class), based on the work of Zhu et al. (2016), to alleviate the problem of unbalancing training data when deriving training samples from the GSPECLib in the Section 3.1, therefore, Figure 8 II and III illustrated that the water body, which was the rare land-cover type in the whole regions, have been accurately captured in the corresponding enlargement figures.

## 5.2 Uncertainty and limitations of the GLC_FCS30-2015 land-cover map

Except for the training sample uncertainties (including sample size, outliers) in the section 5.1, the land-cover heterogeneity also had a significant effect on the classification accuracy (Calderón-Loor et al., 2021; Wang and Liu, 2014). To clarify the relationship between land-cover heterogeneity and overall accuracy of the GLC_FCS30-2015 land-cover map, we firstly used the Shannon entropy to calculate the spatial heterogeneity using the GLC_FCS30_2015 at spatial resolution of $0.05°×0.05°$ (Eq. 4). Figure 11a illustrated the land-cover heterogeneity of GLC_FCS30 land-cover map. Intuitively, the highly heterogeneous regions mainly corresponded to the climatic transition zone especially for the sparse vegetation areas. Then, we combined the land-cover heterogeneity and global validation datasets (in the Section 2.3) to calculate the mean accuracy at different heterogeneity illustrated in Figure 11b. It could be found that the classification accuracy had negative relationship with land-cover heterogeneity with the slope of -0.3347, namely, the GLC_FCS30 had better performance in the homogeneous areas than that of the heterogeneous areas. Similarly, Defourny et al. (2018) also demonstrated that the CCI_LC land-cover products achieved the higher accuracy of 77.36% in the homogeneous areas than that of 75.38% in the all areas.

$$H = -\sum_{i=1}^{n}(P_i \times log_2 P_i) \qquad (4)$$

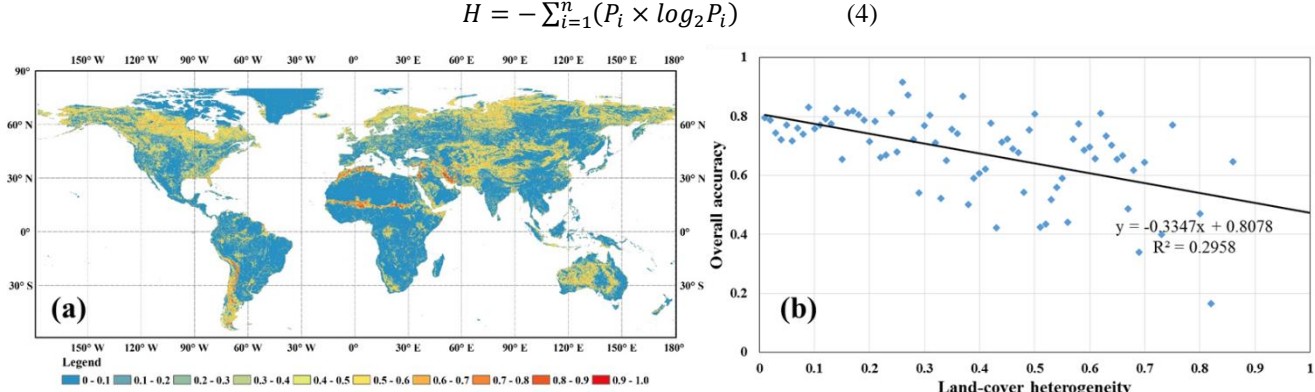

**Figure 11. The land-cover heterogeneity of GLC_FCS30 land-cover map at a spatial resolution of 0.05°, and the relationship between land-cover heterogeneity and overall accuracy using the global validation datasets.**

The CCI_LC map used fine classification system in some region but used coarse classification system in other regions (Defourny et al. 2018). Because the training samples were derived from the CCI_LC land-cover product, our GLC_FCS30 product inherited these characteristics. Therefore, although the GLC_FCS30-2015 provided a global 30-m land-cover product

with 30 land-cover types (Table 2), the 14 LCCS level-2 detailed land-cover types were applied only for certain regions rather than globe, illustrated in the Figure 12. In future work, quantitative retrieval models and multi-source datasets should be combined to improve the diversity of global land-cover types in GLC_FCS30-2015 and further avoid the existence of global LCCS classification system and detailed regional land-cover classification system. This could be done, for example, by using the Fractional Vegetation Cover (FVC) estimation models (Yang et al., 2017a) to retrieve the annual maximum FVC and then distinguish between open and closed broadleaved or needleleaved forests, combining the time-series NDVI to split the evergreen and deciduous shrublands, as well as integrating the GLCNMO training dataset to further distinguish consolidated from unconsolidated bare areas (Tateishi et al., 2014; Tateishi et al., 2011).

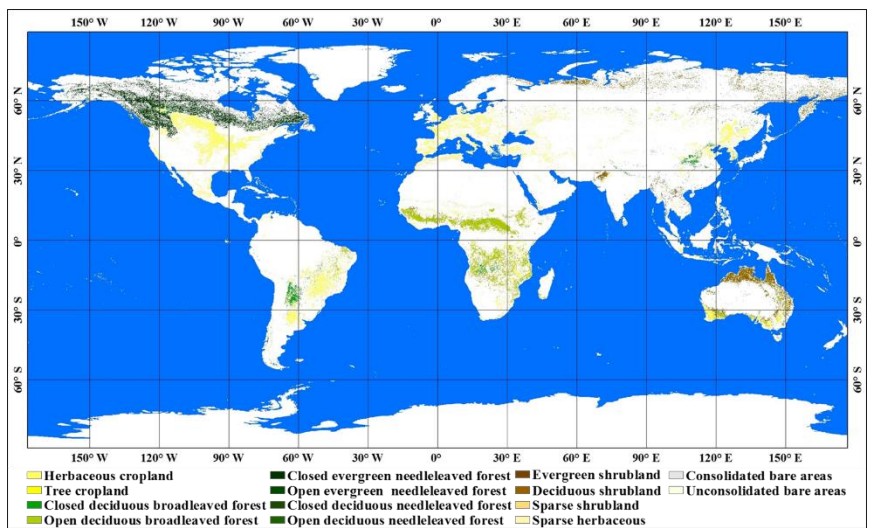

**Figure 12. The spatial distributions of 14 detailed regional land-cover types in the GLC_FCS30-2015 products.**

Due to the differences in classification system, spatial resolution and mapping year, the comparisons between GLC_FCS30-2015 and other land-cover products described in Section 4.2 focused on a qualitative analysis over three regions only. The comparisons illustrated that GLC_FCS30-2015 had great advantages compared to CCI_LC-2015 and MCD12Q1-2015 in terms of spatial detail and had a greater diversity of land-cover types than FROM_GLC-2015 and GlobeLand30-2010; however, quantitative metrics for measuring the advantages and disadvantages of GLC_FCS30-2015 compared to other land-cover types was missing. Therefore, our future work will aim to further optimize the global validation datasets and combine more prior validation datasets so that the performance of these land-cover products can be assessed using common validation data. For example, Yang et al. (2017b) used common validation data to quantitatively assess the accuracy of seven global land-cover datasets over China, and Tsendbazar et al. (2015) analyzed metadata information from 12 existing GLC reference datasets and assessed their characteristics and potential uses in the context of 4 GLC user groups.

## 6 Data availability

The GLC_FCS30-2015 product generated in this paper is available at https://doi.org/10.5281/zenodo.3986871 (Liu et al., 2020). The global land-cover products are grouped by 948 5°×5° regional tiles in the GEOTIFF format, which are named "GLCFCS30_E/W**N/S**.tif", where 'E/W**N/S**' explains the longitude and latitude information of upper left corner of each regional land-cover map. Further, each image contains a land-cover label band ranging from 00-255, and the projection relationship between label values and corresponding land-cover types have been explained in the Table 2 (Section 3.1) and the invalid fill value is labeled as 0 and 250.

The corresponding validation dataset, producing by integrating existing prior datasets, high-resolution Google Earth imagery, time-series of NDVI values for each vegetated point and visual checking by several interpreters, is available at http://doi.org/10.5281/zenodo.3551994 (Liu et al., 2019).

## 7 Conclusion

In this study, a global land-cover product for 2015 that had a fine classification system (containing 16 global LCCS land-cover types as well as 14 detailed and regional land-cover types) and 30-m spatial resolution (GLC_FCS30-2015) was developed by combining time-series of Landsat imagery and global training data derived from multi-source datasets. Specifically, by combining MCD43A4 NBAR, CCI_LC land-cover products and Landsat imagery, the difficulties of collecting sufficient reliable training data were easily solved and the fine classification system was also made use of. Local adaptive random forest models, which allow regional tuning of classification parameters to consider regional characteristics, were applied to combine the time-series of Landsat SR imagery and corresponding training data to produce numerous, accurate regional land-cover maps.

The GLC_FCS30-2015 product was validated using 44,043 validation samples which were generated by combining many prior validation datasets and visual interpretation of high-resolution imagery. The validation results indicated that GLC_FCS30-2015 achieved an overall accuracy of 82.5% and a kappa coefficient of 0.774 for the Level-0 validation system (similar to that of GlobeLand30, which contains 9 major land-cover types), as well as overall accuracies of 71.4% and 68.7% and kappa coefficients of 0.686 and 0.662 for the LCCS level-1 (containing 16 land-cover types) and LCCS level-2 (containing 24 land-cover types) validation systems, respectively. The qualitative comparisons between GLC_FCS30-2015 and other land-cover products (CCI_LC, MCD12Q1, FROM_GLC and GlobeLand30) indicated that GLC_FCS30-2015 had great advantages over CCI_LC-2015 and MCD12Q1-2015 in terms of spatial detail and had a greater diversity of land-cover types than FROM_GLC-2015 and GlobeLand30-2010. The quantitative comparisons against other two 30-m land-cover products (FROM_GLC and GlobeLand30) indicated that GLC_FCS30-2015 achieved the best overall accuracy of 82.5% against FROM_GLC-2015 of 59.1% and GlobeLand30-2010 of 75.9%.Therefore, it was concluded that GLC_FCS30-2015 is a promising accurate land-cover product with a fine classification system and can provide important support for numerous regional or global applications.

**Author contributions.** Conceptualization, Liangyun Liu; Investigation, Xiao Zhang; Methodology, Liangyun Liu and Xiao Zhang; Software, Xiao Zhang and Xidong Chen; Validation, Xiao Zhang, Xidong Chen, Yuan Gao and Jun Mi; Writing – original draft preparation, Xiao Zhang; writing—review and editing, Liangyun Liu.

**Competing interests.** The authors declare that they have no conflict of interest.

**Financial support.** This research was funded by the Strategic Priority Research Program of the Chinese Academy of Sciences (XDA19090200), the Key Research Program of the Chinese Academy of Sciences, grant number ZDRW-ZS-2019-1, and the National Natural Science Foundation of China (41825002).

**Acknowledgments.** We gratefully acknowledge the free access of CCI_LC land-cover products provided by European Space Agency, the MCD12Q1 land-cover products provided by National Aeronautics and Space Administration, the FROM_GLC products provided by Tsinghua University, and the GlobeLand30 land-cover products provided by National Geomatics Center of China.

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
