# Peer review of "GLC\_FCS30: Global land-cover product with fine classification system at 30 m using time-series Landsat imagery"

_Earth System Science Data, 2020_

## Referee Comment (RC1) · Hankui Zhang (Referee) · 18 Dec 2020

It is good to see the study using time series Landsat to map global land cover and making the product public available. The classification legend is finer ($\sim$ 30 classes) than the currently available global 30 m land cover products. The training data are derived from the existing land cover maps (CCI_LC) and the Landsat time series temporal metrics were classified using random forest in the GEE platform. The product is validated using reference data collected from different sources for the validation of the existing land cover products and examined by the authors. Validation showed 82.5% overall accuracy in the 9-class level 0 legend and 68.7% accuracy in the ~30 class

level-2 legend. Furthermore, the authors also make their global validation dataset public available, which could benefit other map producers.

I have a few comments on the clarification of the study. Many sentences are vague including the key information of the methodology.

Issue 1: It is unclear to me whether the training data reflectance comes from MODIS or from Landsat. This is the key of the paper. The term 'Global Spatial Temporal Spectral Library' sounds like the training reflectance is from the MODIS data. If the training data reflectance is derived from MODIS NBAR while the trained model is applied on Landsat surface reflectance, there will be some inconsistencies. Both the Landsat across scene viewing geometry variation and the Landsat and MODIS NBAR solar geometry difference will create inconsistency between MODIS NBAR and Landsat reflectance. MODIS NBAR is defined for local noon solar geometry and the Landsat overpass time is 10:30 am local time. Their solar zenith differences can be up to $20°$ depending on the location and time of the year. Furthermore, there will be spectral band pass difference between the two sensors.

Issue 2: Does the authors imply that the global land cover uses fine classification system in some region but uses coarse classification system in other regions? If so, please make it more explicit in the paper (abstract and conclusion) and clearer (what region uses fine classification system). This is important for users who consider to use the products. What is the CCI_LC coverage?

Issue 3: Something is wrong about no. of classes: "containing 30 land-cover types" and "(24 fine land-cover types)." Later on in Section 3.1, the 34 CCI_LC classes were "removal of four" and "three wetland land-cover types were further combined into one" so there should be 28 classes?

Issue 4: For the level-2 classification legend in Table 2, how the level-1 and level-2 classes can be used together for classification. For example, Deciduous broadleaved forest 60, Closed deciduous broadleaved forest 61, and Open deciduous broadleaved

forest 62 cannot be put together for classification. It is either 60 itself OR both 61 and 62. It cannot be all the three together in classification.

Introduction "stamping effect was noticeable" it is unclear what is stamping effect? Use the term which has been used in the literature.

Figure 1, the Landsat end overlap (row overlaps) cannot be considered as two observations.

Section 2.2 Define what is the GImpS-2015 product.

Section 3.1

This step is not conducted in GEE? The authors stated the Landsat data "were reprojected to the sinusoidal projection of MCD43A4." The "metric centroid" algorithm is proposed in Zhang and Roy 2017 NOT by Roy and Kumar (2016). I don't quite follow what is the purpose of the "metric centroid" algorithm since the training reflectance is derived from MODIS rather than Landsat. The "metric centroid" algorithm is used if the training reflectance is from Landsat and the training class label from MODIS.

3.2 Land-cover classification on the GEE platform Delete the comment on "Hughes phenomenon". Hughes phenomenon is for certain classifiers. I don't think it is still relevant for random forest given large number of training samples. The authors in fact admitted it by saying random forest "is less sensitive to noise and feature selection than other" classifiers.

Section 4 "the yellow marks in Table 5" there is no yellow mark in Table 5. Figure 7. What is the size of figure 7 a, b and c small areas?

Lines 470-475, I would suggest deleting this paragraph. This is a little aggressive.

5 Discussion It is good to see Figure 8. However, it is a little misleading. If Figure 8 only shows the number of training samples, why "where there are relatively uniform land-cover types, there are fewer training samples". I would think the other way around.

For each 5 by 5 degree local training, does the authors also use some training samples outside the 3 by 3 tiles if there is insuffient samples in the 3 by 3 tiles? If so make it clearer in the paper.

"Therefore, it can be assumed that the training data derived from the updated GSPECLib were accurate and suitable for large-area land-cover mapping at 30 m." If the GSPECLib's contribution is only to identify homogenous locations, do not over-emphasize in discussion or conclusion. Use something like derivation of training data from existing land cover products.

Line 525, "applied only for certain regions", which region? Users deserve to know before using the data.

6 Data availability Make it explicit that the validation dataset is also public available. I believe it is an important contribution to the community.

7 Conclusion "global training data derived from GSPECLib". It is a little misleading if the GSPECLib is only to identify homogeneous locations. Use something like derivation of training data from existing land cover products.

---

## Referee Comment (RC2) · Anonymous Referee #2 · 9 Feb 2021

While I appreciate the authors' tremendous efforts in this global-scale mapping project, I have several major concerns. From the remote sensing perspective, the novelty of this project is low. Almost all the methods have been developed and used somewhere in the previous land-cover mapping projects. The classification system proposed in the study looks relatively simple. The study is not targeting the issue - "a fine land-cover system is still lacking" - as described at the end of the Introduction. However, the construction of the training database is a great effort that should be given more emphasis in the description of methods (e.g., adding a flowchart) and in the discussion (e.g., effects of sample outliers on mapping accuracies across land cover classes). See details below. I also feel there is a lack of in-depth discussion. For a large-scale project, data

uncertainties, model calibration, and land-cover heterogeneity could have a significant effect on mapping accuracy. But the current form of discussion is superficial and needs to add a comprehensive evaluation of the developed database.

Detailed comments: Line 10: add 'a' before 'lack'. L15: Include full names with the acronyms when they are first introduced. L99: The "lack of global satellite data coverage" is no longer a challenge for MODIS and Landsat that have been free of charge for over a decade. In fact, we are now in a data-rich era, which is why supercomputing and effective data mining are critical. L145-146: Why not directly using the ASTER GDEM product? The most recent version 3 of GDEM has better accuracy than SRTM. Section 2.3: What are your criteria for deriving how many points for each land cover class? L170: Where did you get the high-resolution imagery? How many points did you check? Following what criteria? L177: great -> big. Section: 3.1: There are multiple steps. I suggest a flowchart to describe your process. Also, how many samples did you collect for the study and for each class? What were your criteria? L214: land-covers -> land cover. L303-304: I do not agree that "classification accuracy was insensitive to these parameters". Please see a review of RF in RS classification by Belgiu and Dragut (2016). L320-322: It is vague how you balanced performance, efficiency, and sample volumes. What criteria did you use? Section 5.1: "huge training samples". Exactly how many samples were used? It is vague to use "exceeded 20 million points". Since building the training sample database is the most important contribution of the project, it is critical and would certainly benefit the users through discussing how the number of the training samples and how sample balance (across classes) have affected the results. The authors lightly touched on the outlier effect, but there is a lack of in-depth analysis and discussion using the data from the present project.

---

## Author Comment (AC2) · 14 Mar 2021

**Response to comments**

**Paper #:** essd-2020-182

**Title: GLC_FCS30:** GLC_FCS30: Global land-cover product with fine classification system at 30 m using time-series Landsat imagery

**Journal**: Earth System Science Data

**Reviewer #2**

While I appreciate the authors' tremendous efforts in this global-scale mapping project, I have several major concerns.

Great thanks for the comment. The manuscript has been improved according to your and another reviewer's comments.

From the remote sensing perspective, the novelty of this project is low. Almost all the methods have been developed and used somewhere in the previous land-cover mapping projects.

Great thanks for the comment. As the global-scale mapping involves tremendous efforts and workloads, we split the project into three parts: 1) the work of "Fine Land-Cover Mapping in China Using Landsat Datacube and an Operational SPECLib-Based Approach" analyzed the accuracy and robustness of the automatic classification strategy; 2) the work of "Development of a global 30 m impervious surface map using multisource and multitemporal remote sensing datasets with the Google Earth Engine platform" used the multi-source and multi-temporal imagery to guarantee the high accuracy of impervious surfaces. 3) Based on our previous works (1-2), we combined the time-series Landsat imagery and GSPECLib to generate the GLC_FCS30-2015 global 30 m land-cover products. Therefore, we think the project is an **incremental innovation** because **the GLC_FCS30-2015 is the first global 30 m land-cover product based on an automatic classification strategy, and has significant advantages over mapping accuracy comparing with the current global 30 m products.**

Zhang, X., Liu, L., Chen, X., Xie, S., and Gao, Y.: Fine Land-Cover Mapping in China Using Landsat Datacube and an Operational SPECLib-Based Approach, Remote Sensing, 11, 1056, https://doi.org/10.3390/rs11091056, 2019.

Zhang, X., Liu, L., Wu, C., Chen, X., Gao, Y., Xie, S., and Zhang, B.: Development of a global 30 m impervious surface map using multisource and multitemporal remote sensing datasets with the Google Earth Engine platform, Earth Syst. Sci. Data, 12, 1625-1648, https://doi.org/10.5194/essd-12-1625-2020, 2020.

The classification system proposed in the study looks relatively simple. The study is not targeting the issue - "a fine land-cover system is still lacking" - as described at the end of the Introduction.

Great thanks for the comment. According to the reviewing in the introduction, the current global 30 m land-cover products mainly used the simple classification system (containing 10 major land-cover types), however, our GLC_FCS30-2015 products adopted the CCI_LC (Climate Change Initiative Global Land Cover) classification system containing 30 land-cover types, so it has significant advantages over land-cover diversity comparing with current global 30 m products (for example, only ten land-cover types in GlobeLand30). Based on the comment, the sentence has been deleted in the Introduction as:

"Overall, due to the difficulties in collecting sufficient accurate training data with a fine classification system and the computing requirements involved, producing a global 30-m land-cover classification with a fine classification system is a challenging and labor-intensive task."

However, the construction of the training database is a great effort that should be given more emphasis in the description of methods (e.g., adding a flowchart) and in the discussion (e.g., effects of sample outliers on mapping accuracies across land cover classes). See details below.

Great thanks for the comment. Based on the suggestion, the details of the deriving training samples have been added (the effects of sample outliers have been explained in the next comment)

[revised manuscript text omitted]

**Detailed comments**

Line 10: add 'a' before 'lack'. L15: Include full names with the acronyms when they are first introduced.

Great thanks for the comment. The missed words were added throughout. The full names of the acronyms (CCI_LC and MCD43A4 NBAR) in L15 have been added.

L99: The "lack of global satellite data coverage" is no longer a challenge for MODIS and Landsat that have been free of charge for over a decade. In fact, we are now in a data-rich era, which is why supercomputing and effective data mining are critical.

Great thanks for the comment. Yes, with the free access of MODIS and Landsat imagery, the "lack of global satellite data coverage" is no longer a challenge. Therefore, the sentence has been revised as:

"Secondly, **the high cost of collecting satellite data with consistent global coverage**, the lack of the high-performance computing requirements and the difficulties in preparing image mosaics also cause problems."

L145-146: Why not directly using the ASTER GDEM product? The most recent version 3 of GDEM has better accuracy than SRTM.

Great thanks for the useful suggestion. The GLC_FCS30-2015 land-cover maps began production in 2019, when GDEM version 3 was not integrated on the GEE platform. Based on your suggestion, our further work would use the GDEM version 3 to replace the SRTM dataset.

Section 2.3: What are your criteria for deriving how many points for each land cover class?

Great thanks for the important comment. The sample size of each land-cover type is determined by the stratified random sampling. The works of Foody et al. (2009) and Olofsson et al. (2014) have detailedly explained how to use the area proportion to calculate the appropriate validation sample size. The part has been added as:

To guarantee the confidence of the validation points, several existing prior datasets (see Table 1), high-resolution Google Earth imagery and time-series of NDVI values for each vegetated point were integrated to derive the global validation datasets. **Many studies have demonstrated that inappropriately sized validation sample could lead to limited and sometimes erroneous assessments of accuracy (Foody et al. 2009 and Olofsson et al. 2014), therefore, a stratified random sampling based on the proportion of the land-cover areas was adapted to determine the sample size of each land-cover type:**

$$n_i = n \times \frac{W_i \times p_i(1-p_i)}{\sum W_i \times p_i(1-p_i)}; \quad n = \frac{(\sum W_i \times \sqrt{S_i(1-S_i)})^2}{[S(\hat{O})]^2 + \sum W_i \times S_i(1-S_i)/N} \approx \left(\frac{\sum W_i S_i}{S(\hat{O})}\right)^2 \quad (1)$$

**where $W_i$ was the area proportion for class $i$ over the globe, $S_i$ is the standard deviation of class $i$, $S(\hat{O})$ is the standard error of the estimated overall accuracy, $p_i$ is the expected accuracy of class $i$ and $n_i$ represents the sample size of the class $i$.**

L170: Where did you get the high-resolution imagery? How many points did you check? Following what criteria?

Great thanks for the comment. The high-resolution imagery came from the Google earth software. There are 22,823 cropland validation samples in the reference dataset have been checked. Lastly, to guarantee the confidence of validation samples, all validation samples were rechecked by three experts using Google Earth software, if the rechecking results of three experts were in disagreement, the cropland point would be discarded. It has been revised as:

**There are 22,823 cropland validation samples in the reference dataset** (Xiong et al., 2017). In addition, due to the possible temporal interval between the acquisition of the reference data and the GLC_FCS30 products (2015), **the reference samples were checked by three interpreters using the high-resolution imagery for 2015 in the Google Earth software, and were discarded if the judgements of three experts were in disagreement. After discarding wrong cropland points and resampling using the formula (1), a total of 6,917 cropland samples in 2015 were retained.**

L177: great -> big.

Great thanks for the comment. It has been corrected.

Section: 3.1: There are multiple steps. I suggest a flowchart to describe your process. Also, how many samples did you collect for the study and for each class? What were your criteria?

Great thanks for the comment. According to your suggestion, the flowchart has been added, and the sample sizes of each land-cover type are calculated by the area proportion. Specifically, the part has been supplemented as:

[Figure]

Figure 3. The flowchart of deriving training samples by using multi-source datasets.

Similar to our previous works (Zhang et al., 2019; Zhang et al., 2018), four key steps were adopted to guarantee the confidence of each training point, as illustrated in the Figure 3. As in Zhang et al. (2019), the spectrally homogeneous MODIS–Landsat areas were firstly identified based on the variance of a 3×3 local window using spectral thresholds of [0.03, 0.03, 0.03, 0.06, 0.03, and 0.03] for the six MODIS bands (blue, green, red, NIR, SWIR1, and SWIR2) in the both MCD43A4 NBAR products and Landsat SR imagery (Feng et al., 2012). It should be noted that the year-composited Landsat SR data were downloaded from GEE platform with the sinusoidal projection. As the MCD43A4 NBAR is corrected for view-angle effects and Landsat has a small view angle of ±7.5°, the view-angle difference between MCD43A4 and Landsat SR could be considered negligible.

Before the process of refinement and labeling, the CCI_LC land-cover products, which had geographical projections, were reprojected to the sinusoidal projection of MCD43A4. The spatial resolution of MCD43A4 was 1.67 times that of the CCI_LC land-cover product and the spectrally homogeneous MODIS–Landsat areas had been identified in the 3×3 local windows. Also, Defourny et al. (2018) and Yang et al. (2017b) found that the CCI_LC performed better over homogeneous areas; therefore, a larger local 5×5 window was applied to the CCI_LC land-cover product to refine and label each spectrally homogeneous MODIS-Landsat pixel. Specifically, the land-cover heterogeneity in the local 5×5 window was calculated as being the percentages of land-cover types occurring within the window (Jokar Arsanjani et al., 2016a). Aware of the possibility of reprojection and classification errors in the CCI_LC products, the land-cover heterogeneity threshold was empirically selected as approximately 0.95; in other words, if the maximum frequency of dominant land-cover types was less than 22 in the 5×5 window, the point was excluded from GSPECLib. After a spatial–spectral filter had been applied to MCD43A4 and a heterogeneity filter to the CCI_LC product, the points that had homogeneous spectra and land-cover types were retained. In addition, to further remove the abnormal points contaminating by classification error in the CCI_LC, the homogeneous points were refined based on their spectral statistics distribution, in which the normal samples would form the peak of the distribution whereas the influenced samples were on the long tail (Zhang et al., 2018). It should be

noted that the geographical coordinates of each homogeneous point were selected as being the center of the local window in the CCI_LC product because this had a higher spatial resolution than that of MCD43A4.

Then, Zhu et al. (2016) and Jin et al. (2014) found that the distribution (proportional to area and equal allocation) and balance of training data had significant impact on classification results, and quantitatively demonstrated that the proportional approach usually achieve higher overall accuracy than the equal allocation distribution. In addition, Zhu et al. (2016) also suggested to extract a minimum of 600 training pixels and a maximum of 8000 training pixels per class for alleviating the problem of unbalancing training data. In this study, the proportional distribution and sample balancing parameters were used to resample these homogeneous points in each GSPECLib 158.85 km×158.85 km geographic grid cell.

L214: land-covers -> land cover.

Great thanks for the comment. It has been corrected.

L303-304: I do not agree that "classification accuracy was insensitive to these parameters". Please see a review of RF in RS classification by Belgiu and Dragut (2016).

Great thanks for the comment. The statement has been revised based on the work of Belgiu and Dragut (2016) as:

"Belgiu et al. (2016) also explained that the classification accuracy was less sensitive to Ntree than to the Mtry parameter, and Mtry was usually set to the square root of the number of input variables. Due to these advantages, the RF classifier is widely used in land-cover mapping"

L320-322: It is vague how you balanced performance, efficiency, and sample volumes. What criteria did you use?

Great thanks for the comment. The reason why we choose the 5°×5° geographical tiles as the mapping unit is because our experiments and the works of Zhang et al. (2017) found that if we chose the 170 km×180 km (the Landsat size) as a spatial unit, there will be lacking of training samples for sparse land-cover types. A good solution is to import some training samples from neighboring 3 by 3 tiles if the training samples are insuffient (Zhang and Roy, 2017; Zhang et al., 2019). Therefore, the 5°×5° geographical tiles, approximately 3×3 Landsat scenes, to avoid the under-fitting when training the local adaptive model. 2) As the GEE has some limitations for computation capability and memory, if we choose bigger spatial unit, the GEE platform would have some over-memory/over-time errors. The sentences have been added as:

"Furthermore, **as illustrated in the previous works, the training samples in a small spatial grid (Landsat scene) were not enough especially for sparse land-cover types, and the training samples from neighboring 3 by 3 tiles were also imported (Zhang and Roy, 2017; Zhang et al., 2019), as well as GEE platform had some limitations for computation capacity and memory**. Therefore,

after balancing the accuracy performance, computation efficiency and training sample volume, the local adaptive random forest models, which split the globe into approximately 948 5°×5° geographical tiles (approximately 3×3 Landsat scenes) similar to our previous work (Zhang et al., 2020), were applied to generate a lot of regional land-cover maps."

Zhang, H. K. and Roy, D. P.: Using the 500 m MODIS land cover product to derive a consistent continental scale 30 m Landsat land cover classification, Remote Sensing of Environment, 197, 15-34, https://doi.org/10.1016/j.rse.2017.05.024, 2017.
Zhang, X., Liu, L., Chen, X., Xie, S., and Gao, Y.: Fine Land-Cover Mapping in China Using Landsat Datacube and an Operational SPECLib-Based Approach, Remote Sensing, 11, 1056, https://doi.org/10.3390/rs11091056, 2019.

Section 5.1: "huge training samples". Exactly how many samples were used? It is vague to use "exceeded 20 million points".

Great thanks for the comment. The exact samples of **27,858,258 points** has been added as:

"In addition, in contrast to other studies that used manual interpretation of samples for global land-cover mapping (Friedl et al., 2010; Gong et al., 2013; Tateishi et al., 2014), the total number of training samples in this study **reaching 27,858,258 points** and so was tens to hundreds of times higher than that used in these global land-cover classifications."

Since building the training sample database is the most important contribution of the project, it is critical and would certainly benefit the users through discussing how the number of the training samples and how sample balance (across classes) have affected the results. The authors lightly touched on the outlier effect, but there is a lack of in-depth analysis and discussion using the data from the present project.

Great thanks for the comment. The effects of training sample sizes and outlier effect have been added in the manuscript in the Discussion Section as:

[revised manuscript text omitted]

---

## Author Response (AR1)

Dear Topical Editor and Reviewers:

On behalf of my co-authors, we thank you very much for reviewing our manuscript and giving us the opportunity to revise the manuscript. We appreciate the comments on our manuscript entitled "GLC\_FCS30: Global land-cover product with fine classification system at 30 m using time-series Landsat imagery" (essd-2020-182).

We have revised the manuscript carefully according to the comments. All the changes were high-lighted (red color) in the manuscript. And the point-by-point response to the comments of the reviewers is also listed below.

Looking forward to hearing from you soon.

Best regards,

Prof. Liangyun Liu

liuly@radi.ac.cn

State Key Laboratory of Remote Sensing, Aerospace Information Research Institute, Chinese Academy of Sciences

No.9 Dengzhuang South Road, Haidian District, Beijing 100094, China

**Response to comments**

Paper #: essd-2020-182

**Title: GLC\_FCS30:** GLC\_FCS30: Global land-cover product with fine classification system at 30 m using time-series Landsat imagery

Journal: Earth System Science Data

**Reviewer** #1**

It is good to see the study using time series Landsat to map global land cover and making the product public available. The classification legend is finer (~30 classes) than the currently available global 30 m land cover products. The training data are derived from the existing land cover maps (CCI\_LC) and the Landsat time series temporal metrics were classified using random forest in the GEE platform. The product is validated using reference data collected from different sources for the validation of the existing land cover products and examined by the authors. Validation showed 82.5% overall accuracy in the 9-class level 0 legend and 68.7% accuracy in the ~30 class level-2 legend. Furthermore, the authors also make their global validation dataset public available, which could benefit other map producers. I have a few comments on the clarification of the study. Many sentences are vague including the key information of the methodology.

Great thanks for the positive comments. The manuscript has been improved according to your and another reviewer's comments.

Issue 1: It is unclear to me whether the training data reflectance comes from MODIS or from Landsat. This is the key of the paper. The term 'Global Spatial Temporal Spectral Library' sounds like the training reflectance is from the MODIS data. If the training data reflectance is derived from MODIS NBAR while the trained model is applied on Landsat surface reflectance, there will be some inconsistencies. Both the Landsat across scene viewing geometry variation and the Landsat and MODIS NBAR solar geometry difference will create inconsistency between MODIS NBAR and Landsat reflectance. MODIS NBAR is defined for local noon solar geometry and the Landsat overpass time is 10:30 am local time. Their solar zenith differences can be up to 20 depending on the location and time of the year. Furthermore, there will be spectral band pass difference between the two sensors.

Great thanks for the key comment. The training data reflectance is derived from Landsat imagery in this study. The MCD43A4 NBAR dataset is used for identifying the spectrally homogeneous MODIS–Landsat areas to further guarantee the confidence of the training data. To make the deriving training samples clearer, the corresponding part has been revised as:

[revised manuscript text omitted]

Issue 2: Does the authors imply that the global land cover uses fine classification system in some region but uses coarse classification system in other regions? If so, please make it more explicit in the paper (abstract and conclusion) and clearer (what region uses fine classification system). This is important for users who consider to use the products. What is the CCI\_LC coverage?

Great thanks for the comment. Yes, as the land-cover labels came from the CCI\_LC products, the GLC\_FCS30-2015 used the level-1 classification system (containing 16 land-cover types) at global scale, and described by a more detailed legend (14 detailed land-cover types) – where available - to reach a higher level of detail in the legend. The spatial distribution of 14 regional and detailed land-cover types has been added in Section 5.2 as:

The CCI\_LC map used fine classification system in some region but used coarse classification system in other regions (Defourny et al. 2018). Because the training samples were derived from the CCI\_LC land-cover product, our GLC\_FCS30 product inherited these characteristics. Therefore, although the GLC\_FCS30-2015 provided a global 30-m land-cover product with 30 land-cover types (Table 2), the 14

LCCS level-2 detailed land-cover types were applied only for certain regions rather than globe, illustrated in the Figure 12.

---

## Author Response (AR2)

Dear Topical Editor and Reviewers:

On behalf of my co-authors, we appreciate the comments on our manuscript entitled "GLC_FCS30: Global land-cover product with fine classification system at 30 m using time-series Landsat imagery" (essd-2020-182).

We have revised the manuscript carefully according to the comments. All the changes were high-lighted (red color) in the manuscript. And the point-by-point response to the comments of the reviewers is also listed below.

Looking forward to hearing from you soon.

Best regards,

Prof. Liangyun Liu

liuly@radi.ac.cn

State Key Laboratory of Remote Sensing, Aerospace Information Research Institute, Chinese Academy of Sciences

No.9 Dengzhuang South Road, Haidian District, Beijing 100094, China

**Response to comments**

**Paper #:** essd-2020-182

**Title: GLC_FCS30:** GLC_FCS30: Global land-cover product with fine classification system at 30 m using time-series Landsat imagery

**Journal: Earth System Science Data**

**(i) Clarity issue:**

The authors replied: "Issue 1: It is unclear to me whether the training data reflectance comes from MODIS or from Landsat. This is the key of the paper. The term 'Global Spatial Temporal Spectral Library' sounds like the training reflectance is from the MODIS data. If the training data reflectance is derived from MODIS NBAR while the trained model is applied on Landsat surface reflectance, there will be some inconsistencies. Both the Landsat across scene viewing geometry variation and the Landsat and MODIS NBAR solar geometry difference will create inconsistency between MODIS NBAR and Landsat reflectance. MODIS NBAR is defined for local noon solar geometry and the Landsat overpass time is 10:30 am local time. Their solar zenith differences can be up to 20 depending on the location and time of the year. Furthermore, there will be spectral band pass difference between the two sensors.

Great thanks for the key comment. The training data reflectance is derived from Landsat imagery in this study. The MCD43A4 NBAR dataset is used for identifying the spectrally homogeneous MODIS–Landsat areas to further guarantee the confidence of the training data." So MCD43A4 is used only for identify homogeneous locations rather than providing any spectra for training.

In this case, please change section 3.1 to avoid misleading:

Change 3.1 Deriving training samples from the GSPECLib-> 3.1 Deriving training samples from CCI_LC

Change the following sentences into one sentence, as most of them are irrelevant:

"As explained in our previous studies (Zhang et al., 2019; Zhang et al., 2018), the Global Spatial Temporal Spectral Library (GSPECLib) was developed to store the reflectance spectra of different land cover types within each 158.85 km×158.85 km geographic grid cell at a temporal resolution of eight days using time-series of the MCD43A4 NBAR and ESA CCI_LC land-cover products. The reasons for selecting the CCI_LC and MCD43A4 NBAR products were that: 1) MODIS has similar spectral bands to the Landsat OLI sensor, and MCD43A4 NBAR has better correction for view-angle effects than other SR products such as MOD09A1, meaning that there is more consistency between MCD43A4 NBAR and Landsat 8 SR (at small view angles, i.e. < 15°) (Feng et al., 2012); and 2) the CCI_LC land-cover product has a detailed classification scheme containing 36 land-cover types, achieves the required classification accuracy over homogeneous areas (75.38% overall), and has a relatively high spatial resolution of 300 m as well as a stable transition between the different annual land-cover products 225 (Defourny et al., 2018; Yang et al., 2017b)."

Change "In contrast to the previous GSPECLib that was used to store the reflectance spectra, the current GSPECLib was developed to derive training samples using the CCI_LC and MCD43A4 NBAR products."->"In contrast to the previous GSPECLib that was used to store the reflectance spectra, in this study GSPECLib was developed to derive training samples locations. The training sample spectra were derived from Landsat data and training sample labels derived from CCI_LC."

Great thanks for the comment. Based on the suggestion, the title of the Section 3.1 has been changed as:

"3.1 Deriving training samples from the CCI_LC land-cover product"

Further, to avoid misleading when deriving training samples, the sentence has been changed as:

"In contrast to the previous GSPECLib that was used to store the reflectance spectra, in this study GSPECLib was developed to derive the location of training samples. The training samples' spectra were derived from Landsat data, while their land-cover labels were derived from CCI_LC."

**(ii) The tile boundary effect**

The authors replied

"For each 5 by 5 degree local training, does the authors also use some training samples outside the 3 by 3 tiles if there is insuffient samples in the 3 by 3 tiles? If so make it clearer in the paper.

Great thanks for the comment. We didn't import the training samples outside the 3 by 3 tiles. Actually, we have built a backup training sample library to avoid missing training samples of sparse land-cover types, however, after using the training samples from neighboring 3 by 3 $5°×5°$ geographical tiles, the missing training samples in the central tile almost were supplemented by neighboring $3 × 3$ tiles, which caused the backup library to lose its function."

Interesting, do the authors find any boundary effect between neighbor tiles? This is a tricky for local training and classification. I understand the authors use 3*3 to classify center tile but still for those inconsistent/transitional area, boundaries effect could be present. Can the authors state in their manuscript whether such boundaries effect was present or not?

Great thanks for the comment. Yes, we also found there was very slight boundary effect between neighbor tiles over transitional areas (such as: bare land transited to sparse vegetation and grassland) because of the spectral similarity after using various local classifiers. Therefore, the problem has been added in the Discussion Section as:

[revised manuscript text omitted]